# Investigation of observed dust trends over the Middle East region in NASA GEOS Earth system model simulations

Adriana Rocha-Lima[1], Peter R. Colarco[2], Anton S. Darmenov[3], Edward P. Nowottnick[4], Arlindo M. da Silva[3], and Luke D. Oman[2]

[1]Physics Department, University of Maryland, Baltimore County (UMBC), Baltimore, MD, USA
[2]Atmospheric Chemistry and Dynamics Laboratory, NASA Goddard Space Flight Center, Greenbelt, MD, USA
[3]Global Modeling and Assimilation Office, NASA Goddard Space Flight Center, Greenbelt, MD, USA
[4]Mesoscale Atmospheric Processes Laboratory, NASA Goddard Space Flight Center, Greenbelt, MD, USA

**Correspondence:** Adriana Rocha-Lima (limadri1@umbc.edu)

**Abstract.** Satellite observations and ground-based measurements have indicated a high variability in the Aerosol Optical Depth (AOD) in the Middle East region in recent decades. In the period that extends from 2003 to 2012, observations show a positive trend of 0.01-0.04 AOD per year or a total increase of 0.1-0.4 per decade. This study aimed to investigate if the observed trend was also captured by the NASA Goddard Earth Observing System (GEOS) Earth system model. To this end, we examined changes in the simulated dust emissions and dust AOD during this period. To understand the factors driving the increase of AOD in this region we also examined meteorological and surface parameters important for dust emissions, such as wind fields and soil moisture. Two GEOS model simulations were used in this study: the Modern-Era Retrospective analysis for Research and Applications, Version 2 (MERRA-2) Reanalysis (with meteorological and aerosol AOD data assimilated) and MERRA-2 GMI Replay (with meteorology constrained by MERRA-2 Reanalysis, but without aerosol assimilation). We did not find notable changes in the modeled 10-meter wind speed and soil moisture. However, analysis of MODIS Normalized Difference Vegetation Index (NDVI) data, did show an average decrease of 8% per year in the region encompassing Syria and Iraq, which prompted us to quantify the effects of vegetation on dust emissions and AOD in the Middle East region. This was done by performing a sensitivity experiment in which we enhanced dust emissions in grid cells where NDVI decreased. The simulation results supported our hypothesis that the loss of vegetation cover and the associated increase of dust emissions over Syria and Iraq can partially explain the increase of AOD downwind. The model simulations indicated dust emissions need to be tenfold larger in those grid cells in order to reproduce the observed AOD and trend in the model.

## 1 Introduction

### 1.1 The Middle East and Climate

The Middle East is defined as the geographical area extending from north-eastern Africa to western Asia, with the Arabian Desert covering most of this region. The climate is hot and dry, with average precipitation of 50 $\mathrm{mm}$ per year in the most arid regions (Saudi Arabia, Qatar, United Arab Emirates, Bahrain) to 700 $\mathrm{mm}$ per year in the wettest regions around Lebanon

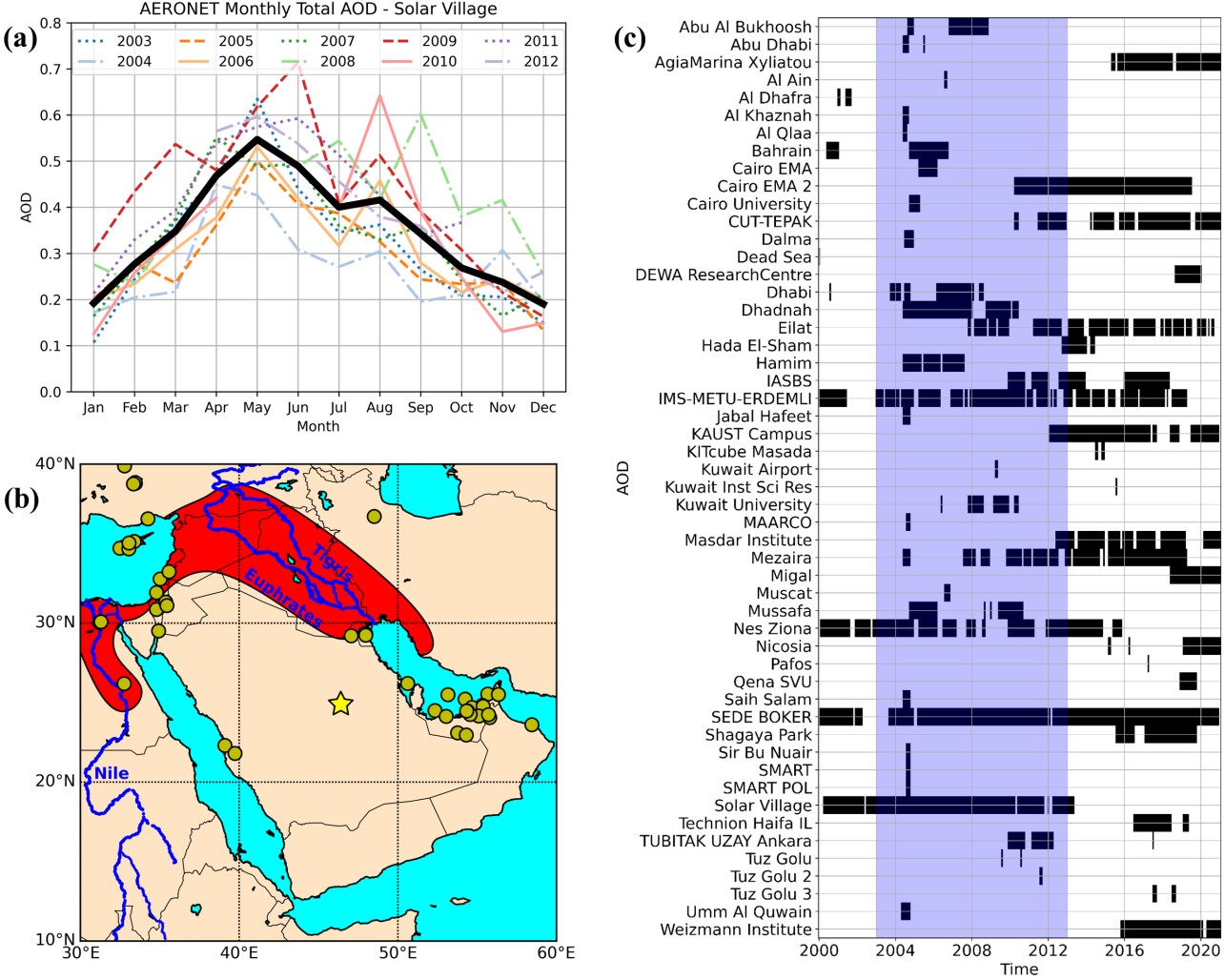

**Figure 1.** a) Aerosol Robotic Network (AERONET) Monthly Total AOD (Level 2.0) product for Solar Village at Saudi Arabia. b) Locations of AERONET stations (yellow circles) in the Middle East region. Red shading corresponds to the Fertile Crescent region. Solar Village is shown in yellow star. c) Availability of AOD data from the AERONET stations. The period of interest for this study (2003-2012) is highlighted.

(Hasanean and Almazroui, 2015), with the driest areas facing more than 300 dry days per year to up to a full year of dry days in desert areas of Saudi Arabia and southern Iraq (Lelieveld et al., 2012). The dust seasonal cycle in the Middle East is different depending on the region (Rezazadeh et al., 2013). Typically, higher aerosol levels occur in the drier season between March and September (see Fig. 1a). Intense dust storms in the Middle East are caused by thunderstorm cells or else by haboobs, which are characterized by walls of blowing sand and dust carried to higher altitudes by strong winds and are common in many parts

of the region (Miller et al., 2008; Knippertz et al., 2007). Dust storms have numerous adverse effects - disruption of roads and flight cancellations, damage to crops, and soil degradation. Populations exposed to high dust concentration are subject to respiratory health problems (Middleton, 2017; Meo et al., 2013). Moreover, airborne dust particles can spread diseases by carrying bacteria, viruses, and pollutants (Morris et al., 2011).

Long-term measurements of aerosol concentrations are scarce over desert areas. The Aerosol Robotic Network (AERONET) (Holben et al., 1998) provides ground-based sun photometer measurements of the column integrated total Aerosol Optical Depth (AOD). Figures 1b-c show AERONET stations located in the Middle East region. Stations are mainly concentrated near populated coastal areas, and only a few of them have long-term consistent high quality AOD (referred to as Level 2.0) time-series extending over the last 20 years (Sede Boker, Solar Village, Nes Ziona, and IMS Metu Erdemli), with Solar Village being the only station in the central Middle East area. Estimates of total dust emissions are poorly constrained, and the locations of the dust sources not well known, with an estimated contribution of about 75% from natural and 25% from anthropogenic sources (WorldBank, 2019). According to Kok et al. (2021), global annual dust emissions from the Middle East and central Asia contribute 30% of the total world global dust loading in the atmosphere.

In recent decades, observations have shown that some regions in North Africa and the Middle East have experienced an overall increase in the frequency and intensity of dust storms, whereas other regions have experienced a decrease (Shao et al., 2013). Investigation of global AOD trends prior to 2010 using model simulations concluded that models underestimate changes over this region (Chin et al., 2014; Pozzer et al., 2015). According to Hamidi et al. (2013), dust activities in the years preceding 2013 were intensified due to several reasons. These include the development of dam construction projects on the Tigris and Euphrates rivers, which decreased the water content of soil in the downstream areas, urbanization in regions previously used for agriculture, and a shortage of power that hindered the adequate irrigation of farmlands. The meteorological factors behind these trends are not well understood (Albugami et al., 2019). Notaro et al. (2015) linked the variability in the dust activity in the Arabian peninsula with prolonged drought in the Fertile Crescent region (the region that extends from Nile river in Egypt to the nearby Tigris and Euphrates rivers - see Fig. 1b). Using remote sensing observations, Nabavi et al. (2016) identified the region northwest of Iraq and east of Syria as emerging dust areas with a marked increase in the frequency of dust events. Xia et al. (2022) showed that the Middle East AOD trend has reverted in the most recent years partially due to expansion of irrigated areas. Conversely, Che et al. (2019) found that sea level pressure and wind speed were the primary meteorological factors driving AOD variations over the Middle East. More recent studies have examined the link between dust activity in the Middle East and climate decadal oscillations. Xi (2021) associated the AOD trends over the Middle East with the combined effects from El Niño-Southern Oscillation (ENSO) and the Pacific Decadal Oscillations (PDO). Specifically, when both ENSO and PDO are in phase, influences in the sea surface temperature and winds are amplified, creating high surface pressure around the Middle East with hotter and drier conditions. The drought in the Tigris-Euphrates basin is believed to be associated with effects of La Niña and negative PDO phases, which can have resulting effects for agriculture and vegetation loss in the Fertile Crescent. At the same time, Liu et al. (2023) related the shift of the AOD trend in the Middle East around 2010 from positive to negative to the shift in the North Tropical Atlantic (NTA) Sea Surface Temperature (SST).

Klingmüller et al. (2016) investigated the increase of dust emissions in the Middle East and showed that observations from AERONET at Solar Village have a positive trend in the period between 2003 and 2012 (Fig. 2a). After 2012, AERONET data shows that AOD started to decrease again; unfortunately, reliable measurements are not available at that station beyond May 2013. Klingmüller et al. (2016) also showed that AOD from the space-based Moderate-resolution Imaging Spectrora-

diometer (MODIS) confirmed a positive trend in the total AOD over Solar Village and other areas across the Middle East region in the period between 2003 and 2012, with a reverse of the trend until 2015 (i.e., the end of their study).

In this study, we evaluated the ability of the NASA Goddard Earth Observing System (GEOS) (Rienecker et al., 2008; Molod et al., 2015) a global Earth system model developed by the NASA Global Modeling and Assimilation Office (GMAO), to reproduce the observed dust AOD trend between 2003-2012. For that, we look at the variations in dust emission, dust AOD,

and changes in the main dust driven meteorological parameters in the same period when observations indicated a positive AOD trend. Two model simulations based on the same GEOS model version were used in this study: the Modern-Era Retrospective analysis for Research and Applications, Version 2 (MERRA-2) Reanalysis (with meteorological and aerosol AOD data assimilated) and the MERRA-2 GMI Replay (with meteorology constrained by MERRA-2 Reanalysis, but without aerosol assimilation). We investigated the possible causes of the AOD variability and the qualitative impact of the change in vegetation

cover by performing a sensitivity study to allow for an increase of emissions over areas of decreasing Normalized Difference Vegetation Index (NDVI) in the region of the Fertile Crescent.

## 1.2 Model Description and Dust Emission Scheme

The GEOS model is a global Earth system model that supports NASA's broad range of Earth science applications, including data analysis, reanalysis, observing system simulation experiments, climate and weather prediction, and basic Earth system

research. The version of GEOS used in this study includes prognostic aerosols from the Goddard Chemistry, Aerosol, Radiation, and Transport (GOCART) module (Chin et al., 2002; Colarco et al., 2010). The GOCART aerosol module includes the sources, sinks, and chemistry of dust, sulfate ($SO_4$), sea salt, and black (BC) and organic carbon (OC) aerosols. For dust, the particle size distribution is discretized into five non-interacting size bins spaced between 0.1-10 µm radius.

Dust emissions are known to be sensitive to different factors, climatology, wind patterns, topography, and soil characteris-

tics. In the default configuration of GEOS, the description of the spatial distribution of dust emission is parametrized by the topographic source based on Ginoux et al. (2001). The uplifting of the dust particles is modulated by the wind fields and soil wetness according to the dust flux equation:

$$F_p = \begin{cases} C_g \cdot S \cdot s_p \cdot U_{10}^2 \cdot (U_{10} - U_t), & U_{10} > U_t \\ 0, & \text{otherwise} \end{cases} \tag{1}$$

where $C_g$ is a global tuning constant, $S$ is the emission efficiency given by the topographic source function, $s_p$ is the mass

fraction of dust for each size bin ($p$), $U_{10}$ is the horizontal wind speed at 10 meter, and $U_t$ is the threshold wind speed dependent on the particle size and soil moisture (volume of water within the volume of bulk soil) required to initiate emission (Belly,

1964). Ginoux et al. (2001) replaced the soil moisture variable by the soil surface wetness, a dimensionless variable ranging from zero to one that indicates the saturation level of the soil. Soil surface wetness above 0.5 results a complete saturation of the soil (effectively $U_t \to \infty$) and zero flux emission.

Two model GEOS simulations were used in this study: MERRA-2 Reanalysis and MERRA-2 GMI Replay. MERRA-2 Reanalysis is a long-term (1980-present) global reanalysis that assimilates satellite meteorological and aerosol data (Gelaro et al., 2017; Randles et al., 2017; Buchard et al., 2017). It assimilates several wind observations, including ground-based datasets, remotely sensed profilers, and satellite derived and retrieved winds (McCarty et al., 2016). MERRA-2 Reanalysis also uses precipitation observations to correct model-generated precipitation, which is needed for estimating soil moisture in the
catchment land surface model (De Lannoy et al., 2014; Gelaro et al., 2017; Reichle et al., 2017a, b). Furthermore, MERRA-2 Reanalysis assimilates total AOD from multiple systems, such as the Advanced Very-High-Resolution Radiometer (AVHRR), MODIS, Multi-angle Imaging Spectroradiometer (MISR) over bright surfaces, and from selected AERONET stations prior to 2015. The aerosol assimilation is performed eight times a day at three-hour intervals. MERRA-2 GMI Replay (Strode et al., 2019) uses a replay mechanism to produce a simulation with meteorology similar to MERRA-2 Reanalysis. There are
several differences between MERRA-2 GMI Replay and MERRA-2 Reanalysis to note. First, unlike MERRA-2 Reanalysis, MERRA-2 GMI Replay was performed with a full chemistry simulation using the Global Modeling Initiative's (GMI) chemical mechanism (Duncan et al., 2007; Strahan et al., 2007). This has no practical impact on the simulations of dust emissions and loss processes. Second, MERRA-2 GMI Replay, uses the same catchment land surface model as MERRA-2 Reanalysis, however, there are differences in soil moisture due to differences in the treatment of water vapor and precipitation. Finally, and
most significantly, MERRA-2 GMI Replay does not constrain the total aerosol optical depth to observations like in MERRA-2 Reanalysis.

Both MERRA-2 and MERRA-2 GMI were run at a global ~50 km horizontal resolution with 72 vertical levels extending from the surface to ~80 km altitude. Output is saved hourly on a regular grid with resolution of 0.625° longitude and 0.5° latitude (Gelaro et al., 2017).

**2   Comparison of Dust AOD Trends between Observations and GEOS Model**

Using the same definition given by Klingmüller et al. (2016), we calculated the deseasonalized AOD for our model simulations and various remote sensing aerosol observations as the difference between monthly mean AOD and the average monthly mean for the period 2003-2012. The slope of the deseasonalized AOD for the same period was obtained by fitting a linear regression to the deseasonalized AOD. Table 1 summarizes ground-based and satellite products used in this study. All datasets were
selected to cover the period between 2003 and 2012.

Figure 2 shows the slope of the deseasonalized AOD at Solar Village in Saudi Arabia for: a) AERONET-Level 2.0, b) MODIS Deep Blue Collection 6.1 (MODIS DB C6.1), c) MISR, d) MERRA-2 Reanalysis, and e) MERRA-2 GMI Replay. Observations from AERONET and MISR show positive slopes of 0.017 and 0.011 per year respectively. For MODIS, a positive slope of 0.005 per year is obtained averaging the deseasonalized AOD of nine grid cells around Solar Village, shown as the

**Table 1.** List of sensors and dataset products used in this study in the period from 2003 to 2012.

| Sensor | Dataset Description |
| --- | --- |
| AERONET Sun-photometer | AOD Level 2.0, Solar Village (24.9N, 46.4E) (Holben et al., 1998) |
| MISR Terra Satellite | AOD Level 2 Aerosol (NASA/LARC/SD/ASDC, 1999) |
| MODIS Terra Satellite | AOD Deep Blue Collection 6.1 (Hsu et al., 2019; Sayer et al., 2019) |
| MODIS Aqua Satellite | Normalized Difference Vegetation Index (NDVI) MYD13C2 Version 6, Level 3 product, 0.05 degree (Didan, 2015) |
| MODIS Terra/Aqua Satellite | Global Daily Terrestrial Gross Primary Production (FluxSat GPP) Version 2.0 (Joiner and Yoshida, 2020) |

squared area in Fig. 3a. Although the slope of the AOD from MODIS DB C6.1 at these grid cells around Solar Village has a *p*-value equal 0.08, incompatible to a trend (typically *p*-values lower than 0.05 is statistically significant), that is not the case for the next neighboring grid cells, as will be shown in Fig. 3a. At Solar Village grid box, MERRA-2 Reanalysis shows a significant slope of increase of AOD of 0.012 per year and MERRA-2 GMI Replay does not have any trend, as confirmed by *p*-values on the order of $10^{-8}$ and 0.11 for each fitting, respectively.

To further evaluate the model capability to capture dust AOD trends over the region, MODIS AOD was "dust screened" to select AOD observations consisting predominantly of dust aerosol. For that, we selected observations with an Angstrom exponent smaller than 1.0 and with Single Scattering Albedo (SSA) at 670 nm larger than at 412 nm. Then, AOD from MERRA-2 Renalysis and MERRA-2 GMI were sampled at MODIS dust screened time. Figure 3 shows the regional map of the slope of the dust screened deseasonalized AOD (on the left) for the period between 2003 to 2012, and respective *p*-values

for the linear fitting of the slope (on the right). MODIS and MISR show similar spatial trends extending from the region of the Fertile Crescent in Syria and Iraq around the Tigris and Euphrates rivers to the south border of Saudi Arabia, with positive slopes varying from 0.01 to 0.04 per year. MISR shows higher slope values in the southeast border of the region near Oman and the United Arab Emirates. Notably, the higher slope hotspots both in MODIS and MISR have statistically significant *p*-values. The MERRA-2 Reanalysis shows a positive variability in the AOD similar to MODIS and MISR, although the variation differs

in the magnitude and does not capture the hotspot region over the Fertile Crescent. The MERRA-2 GMI Replay does not show any significant variability in the same period at Solar Village. The white pixels patches correspond to grid boxes with low statistics (less than 50 observations in average per month).

## 3   Investigating causes of the AOD variability

Dust emissions in both MERRA-2 Reanalysis and MERRA-2 GMI Replay are driven by similar wind fields, however they

used different soil moisture datasets. Dust emissions are modulated by the topographic source based on Ginoux et al. (2001).

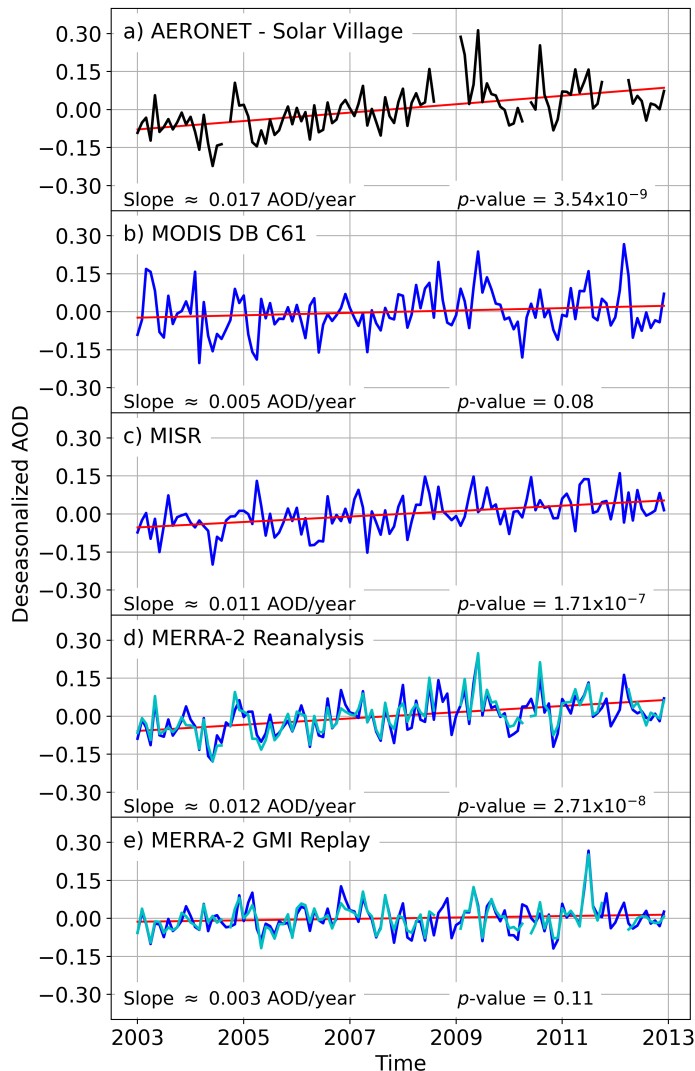

**Figure 2.** Time-series of the deseasonalized AOD at Solar Village in Saudi Arabia for a) AERONET ground based station level 2.0, b) MODIS Deep Blue Collection 6.1 (C6.1) (Hsu et al., 2019; Sayer et al., 2019), and c) MISR (NASA/LARC/SD/ASDC, 1999) Simulated time-series of the deseasonalized AOD for d) the MERRA-2 Reanalysis (i.e., with aerosol data assimilated), and e) MERRA-2 GMI Replay (no aerosol data assimilated). The lines in light blue color overlapped in the simulation plots correspond to the same model datasets sampled at AERONET time.

Figure 4 shows the deseasonalized slopes of the 10-meter wind speed, surface soil wetness, and dust emissions fluxes for both MERRA-2 Reanalysis and MERRA-2 GMI, as well the corresponding *p*-values.

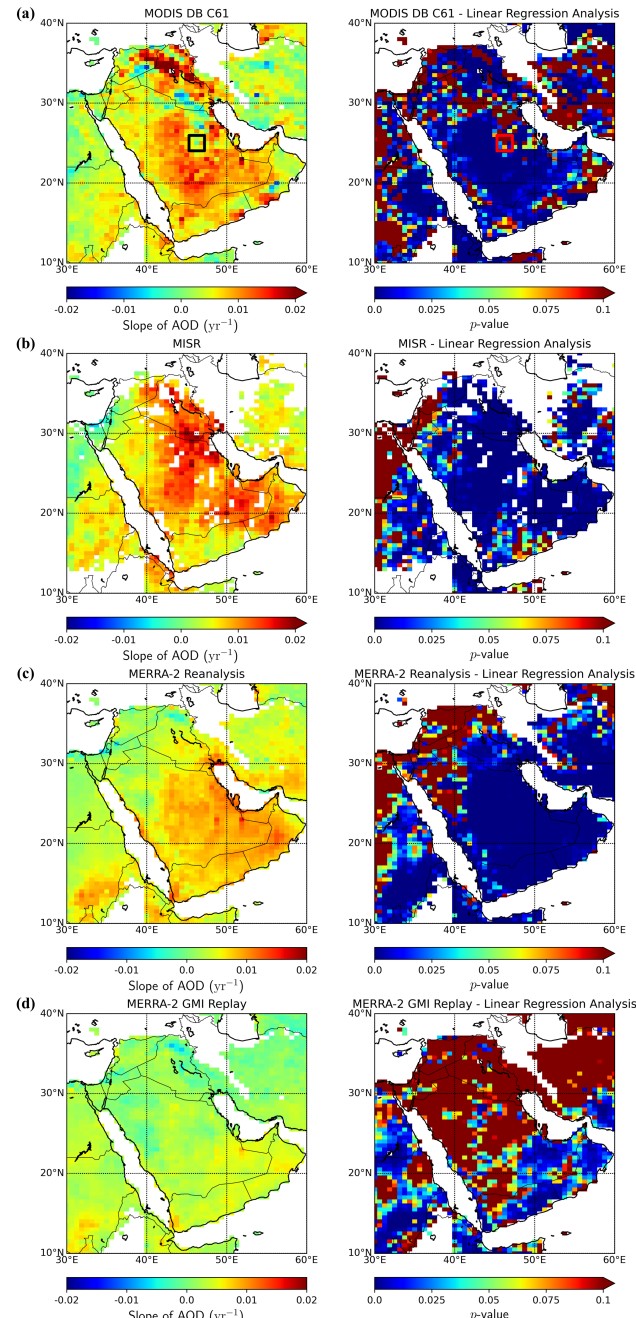

**Figure 3.** Map of the dust screened deseasonalized AOD (on the left) for the period of 2003-2012, and *p*-values of the linear regression (on the right): a) MODIS Deep Blue Collection 6.1, b) MISR, c) MERRA-2 Reanalysis, and d) MERRA-2 GMI Replay. GEOS model simulations datasets are synchronized to match MODIS DB C6.1 time.

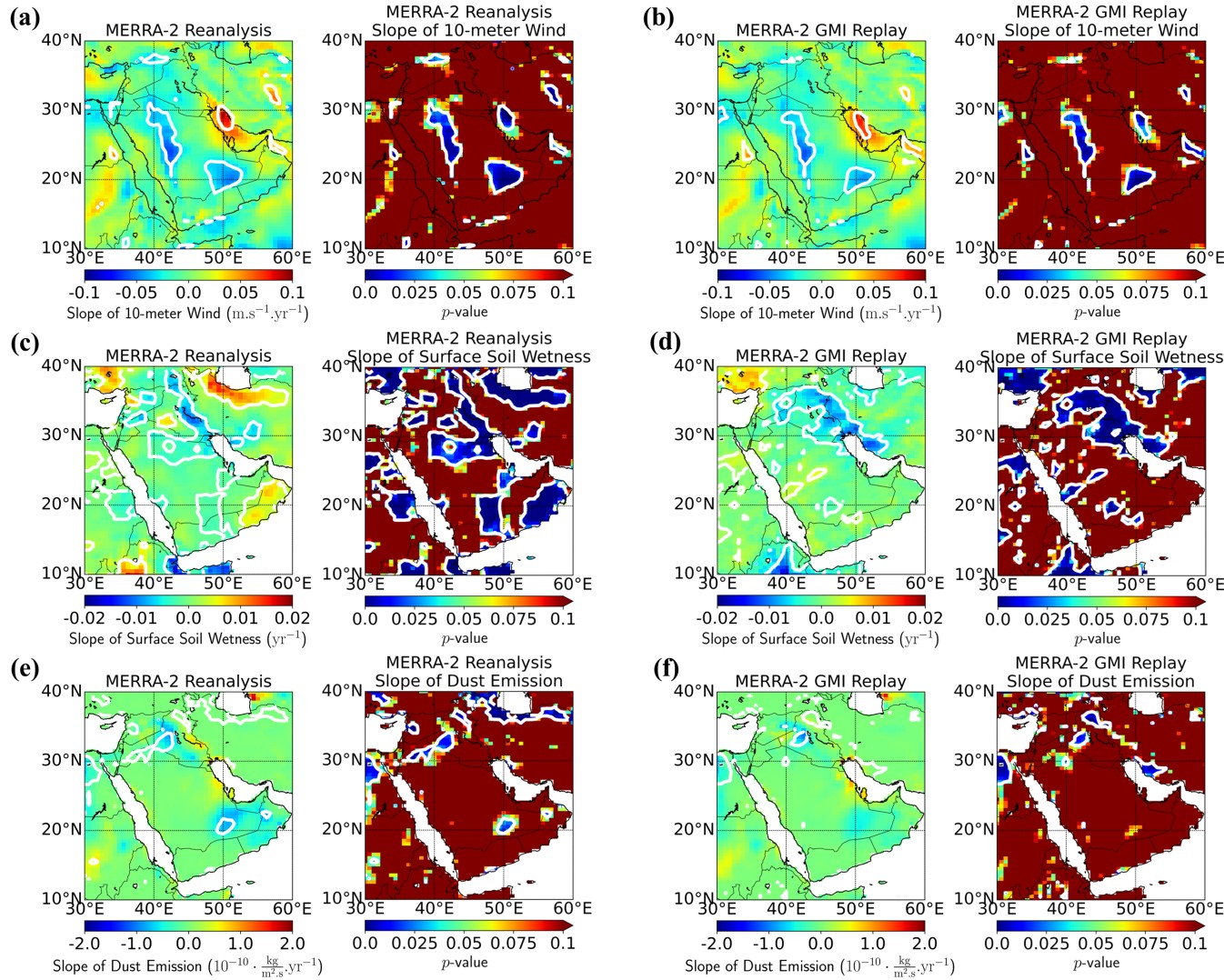

**Figure 4.** MERRA-2 Reanalysis (two columns on the left) and MERRA-2 GMI (two columns on the right): a-b) slope of the 10-meter wind speed, c-d) slope of surface soil wetness, and e-f) slope of dust emissions. All slopes were calculated for the period between 2003 and 2012. White contour regions with *p*-values lower than 0.05 are associated with statistically significant slopes.

The slopes of the 10-meter wind fields (Fig. 4a-b) for both MERRA-2 Reanalysis and MERRA-2 GMI Replay have corresponding $p$-values that are statistically significant only over a few regions (shown in blue), and in those areas the slope of the surface wind is negative over land. This would suggest that the changes in the 10-meter wind would result in weaker regional dust emissions in the model. The greatest magnitude change in 10-meter wind speed over these regions is on the order of $-0.06 \ \mathrm{m \cdot s^{-1} \cdot yr^{-1}}$, which comparatively to calculated 10-meter annual mean baseline values for the region, represents changes of less than 4% (see Appendix A for baseline values).

For the surface soil wetness, we observe that the slopes for MERRA-2 Reanalysis and MERRA-2 GMI have different spatial distribution, which we attributed to the differences in the soil moisture inputs used in them. For MERRA-2 Reanalysis, areas with statistical significance are observed over Oman, with a positive variation of up to $+0.0075 \ \mathrm{yr^{-1}}$, and over Iraq and Syria, with a negative variability of up to $-0.01 \ \mathrm{yr^{-1}}$. Interestingly, the region of the Fertile Crescent shows a decrease in the surface soil wetness for both simulations. For MERRA-2 GMI Replay, where we do not observe the increase in surface soil wetness over Oman, the most significant change in soil moisture is seen over Iraq and along the left border of Iran with decrease in soil moisture of up to $-0.0075 \ \mathrm{yr^{-1}}$ and in central areas of Saudi Arabia with positive increase of $+0.005 \ \mathrm{yr^{-1}}$.

Although variations in the surface soil wetness is observed between MERRA-2 Reanalysis and MERRA-2 GMI Replay, the slope of the dust emissions fluxes are essentially identical in both reanalysis and replay (Fig. 4e-f). Only a few regions show $p$-values with statistical significance. In those few spots, the model indicates: (1) decrease in dust emissions correlated to the decrease in wind speeds to the south of Saudi Arabia and (2) slight decrease (increase) in dust emissions correlated to increase (decrease) in soil moisture. Overall, we observe that there is no statistically significant increase in dust emissions. It also confirms that the positive slope in the deseasonalized AOD in the MERRA-2 Reanalysis cannot be explained by the local increase in dust emission and it was partially captured only due to the AOD assimilation. Moreover, the few locations of statistical significance in the slope of the dust emission are places where the emission efficiency given by the topographic source (see Fig. 7a) is relatively low, which can also be seen in the baseline values for monthly and annual dust emission shown in Appendix A, Fig. A4 and Fig. A7.

To explore the impact of aerosol data assimilation on the MERRA-2 simulation, Fig. 5 shows the slope of the deseasonalized AOD analysis increments from MERRA-2. The aerosol analysis in MERRA-2 is performed by combining the forecasted column integrated AOD with the assimilated AOD increments. Quality-controlled AOD at 550 nm is assimilated into the GOCART and GEOS model by the Goddard Aerosol Assimilation System (GAAS) every 3 hours (Buchard et al., 2015, 2016). For desert regions in the period between 2003-2012, AOD was assimilated using observations from MISR and AERONET AOD Level 2.0 (Randles et al., 2017). Figure 5a-b shows that the deseasonalized slope of the assimilated AOD increments is positive over the entire Middle East region, with two main hotspots over Iraq and Solar Village in Saudi Arabia, consistent with the observations (Fig. 3a-b). Significant $p$-values were obtained over the entire region confirming the positive AOD trend over the period. These results show that data assimilated from the Solar Village ground based station is driving the increase more so than assimilation of spaceborne observations. In the MERRA-2 Reanalysis (Fig. 3c), the resulting AOD increase will be seen downwind due to the typical configuration of the wind patterns over the region in the direction north-south. The wind points in the direction to transport dust from the Fertile Crescent to the Saudi Arabia region (Fig. 5b).

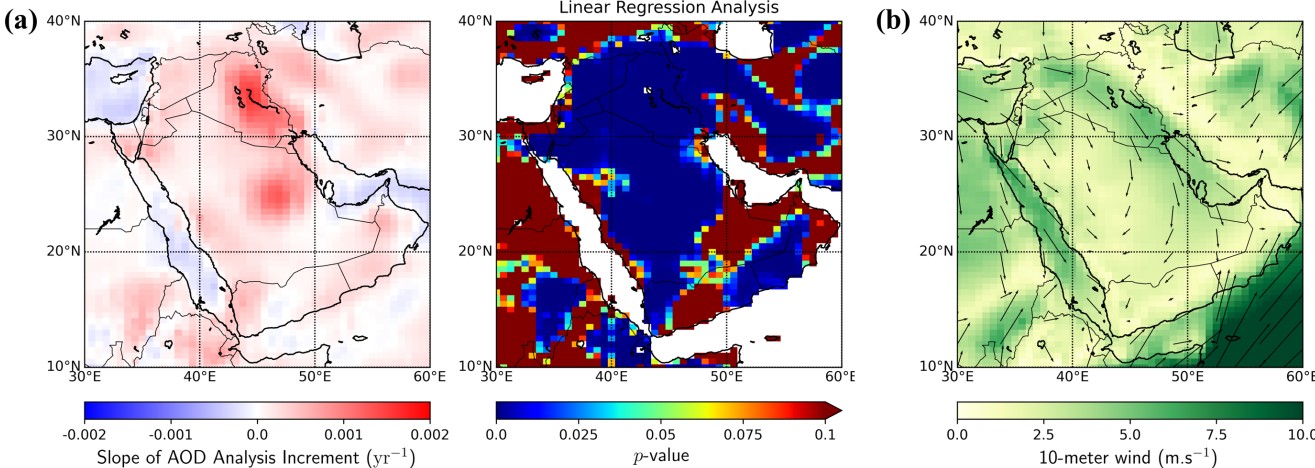

**Figure 5.** a) Slope of the AOD Analysis Increments assimilated in MERRA-2 Reanalysis in the period between 2003-2012 and *p*-values of the linear regression show an overall positive increment over the entire Middle East region and b) direction and magnitude of 10-meter wind vectors for the month of June over the Middle East region.

## 3.1 Investigating Causes of the AOD Variability over the Middle East Region

### 3.1.1 Normalized Difference Vegetation Index (NDVI)

The causes of recent AOD trends over the Middle East have not been clearly identified yet. Although aerosol emissions from anthropogenic sources have increased over some regions, Klingmüller et al. (2016) showed that changes in AOD in the same period were accompanied by decreases in the MODIS Angstrom exponent and so decreases in the AERONET fine mode particles. Similarly, Sabetghadam et al. (2021) noticed a higher presence of aerosol dust type in March 2012 compared to the long-term average for the period of 2001-2019 in the Fertile Crescent region. These findings suggest that the AOD trends over

the region is associated to higher concentration of coarse particles, like dust. Increases in soil dryness due the land use and higher temperatures have also been linked to the observed AOD trends (Adamo et al., 2022). Changes in the vegetation cover shows reduction of vegetation in the region of the Fertile Crescent as seen in Fig. 6a.

The Normalized Difference Vegetation Index (NDVI) dataset product from MODIS provides a consistent global spatial temporal comparison of canopy greenness (Didan, 2015). Figure 6a shows the time series of the monthly NDVI for a grid box

located in the region of Fertile Crescent. The greenest peak in March-April is suppressed over the years, with NDVI values ranging from 0.4 (sparsely vegetated) in the year 2003 to 0.1 (bare ground) in 2012. NDVI maps for the entire region are shown in Fig. A8. Figure 6b-c shows the map of the slope of the deseasonalized NDVI and respective *p*-values between 2003-2012, confirming the decrease of vegetation in the Fertile Crescent region. Figure 6d shows the annual variation of the NDVI in percentage obtained as the slope divided by the annual baseline of the NDVI over the region, resulting in maximum NDVI

variations up to around 80% over the decade in the most critical grid box.

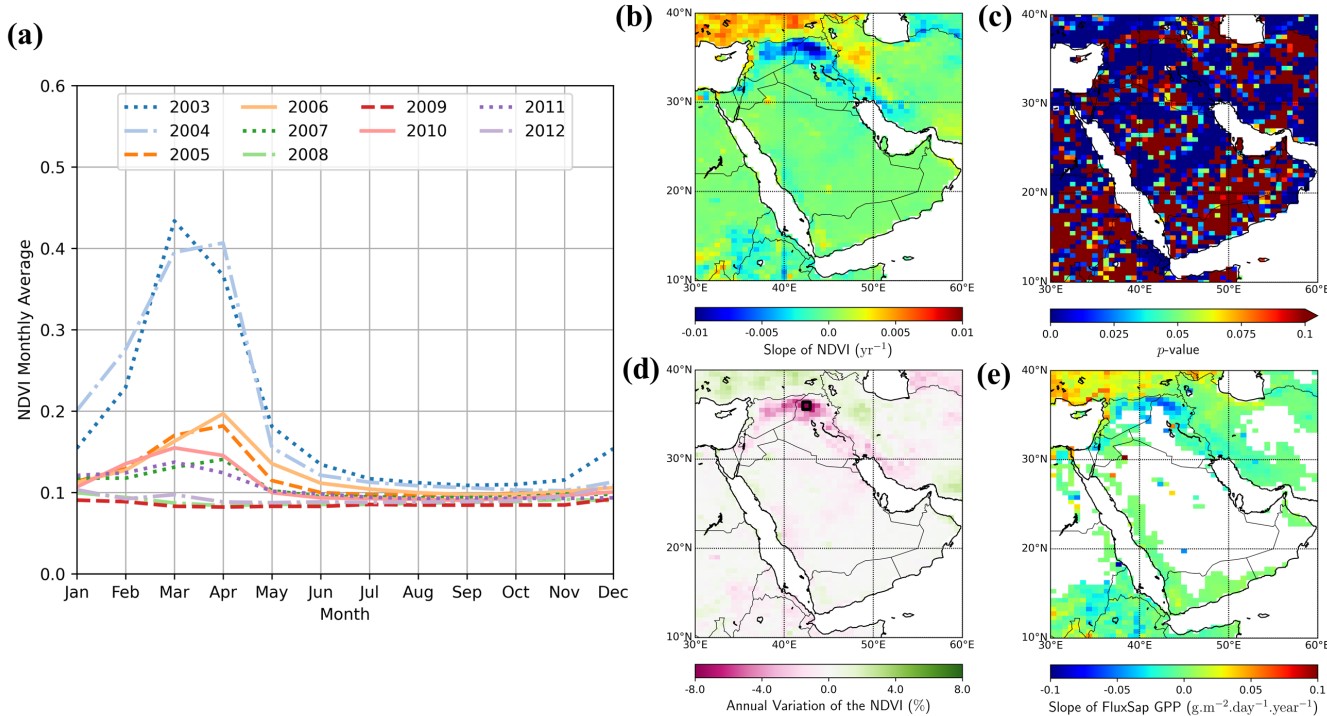

**Figure 6.** a) Monthly NDVI product from MODIS between 2003 and 2012 for a grid cell in the region of Fertile Crescent (square in Fig. 6d), b) slope of the deseasonalized NDVI for the same period, c) *p*-values of the linear regression fitting of the slope of the NDVI, d) equivalent annual variation calculated as the magnitude of the slope divided by the annual mean NDVI in percentage, and e) slope of the Global Daily Terrestrial Gross Primary Production (FluxSat GPP) obtained for the same period.

The determination of vegetation index is usually associated with high uncertainty, as retrievals of NDVI over this region can be influenced by dust aerosols given that surface reflectance used to compute NDVI can be affected by absorbing aerosols. Different sensors or algorithms have shown consistent results indicating the existence of deforestation over this region. The Gross Primary Production (GPP), which represents the amount of carbon dioxide ($CO_2$) assimilated by plants through pho-
tosynthesis, is an important indicator of vegetation. The GPP product archived at the NASA Aura Validation Data Center (AVDC) was derived using neural networks combined with Bidirectional Reflectance Distribution Function (BRDF) and Nadir Adjusted Reflectance (NBAR) products from the MODIS instrument. It also incorporates global GPP estimates from selected FLUXNET2015 eddy covariance tower sites (Joiner and Yoshida, 2020). Figure 6e shows the slope for MODIS and FLUXNET-derived Global Daily Terrestrial Gross Primary Production (FluxSat GPP) product. In this product, productivity of biomass is
expressed in units of biomass carbon $(g \cdot m^{-2} \cdot day^{-1})$, which is similar to NDVI index. The negative slope for the productivity biomass over the Fertile Crescent is an additional indicator of decrease in vegetation in that region.

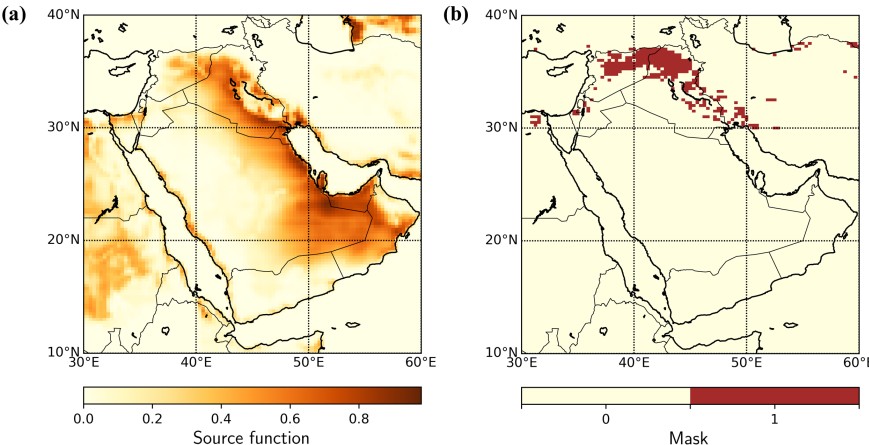

**Figure 7.** a) Topographic source function shows the spatial efficiency of dust emission over the Middle East region based on Ginoux et al. (2001, 2004). The map is based on observations that topographic lows with bare soil have accumulated sediments and are potential dust sources and b) mask used for simulation of dust emissions in the region of the Fertile Crescent coincident with grid boxes in the model where decreasing vegetation coverage was observed to have negative slopes of NDVI above 0.025 $\mathrm{yr}^{-1}$.

## 3.2 Model Simulation to Test Effects of Enhancing Dust Emissions over Areas with Decreasing Vegetation

Changes in vegetation coverage were not directly taken into account in the version of the GEOS model in which MERRA-2 Reanalysis and MERRA-2 GMI Replay are based, given that the efficiency of emission is specified by the topographic source function that is static over time ($S$ in Equation 1). Additionally, as seen in Fig. 4a-d, variations in the 10-meter wind speed and soil moisture in the model were mostly negligible over the region, and indeed the impact of those changes is revealed as having only a small impact on the computed dust emissions (Fig. 4e-f). For the few locations where statistically significant changes in dust emission happened, they were mostly in the direction to decrease the emissions. In the region where the changes were in the direction to increase emissions, it did not have a great impact.

A study by Kim et al. (2013) showed that the time dependence of global dust sources can have significant impacts on dust simulations near source regions. However, the static topographic source function used in MERRA-2 Reanalysis and MERRA-2 GMI Replay does not incorporate time-varying NDVI. To assess the impact of the variation of vegetation on dust emissions, we performed a GEOS simulation (baseline case) using the default static topographic depression source map (Fig. 7a) and a second simulation with a modified source (dust enhanced case). The mask (Fig. 7b) was obtained by selecting grid cells with NDVI variations smaller than $-0.0025 \ \mathrm{yr}^{-1}$. The apportioned model run with the mask allowed the simulation of the dust emissions only in the region where observations indicate desertification. Both simulations (baseline and dust enhanced case) were performed with a similar configuration and were replayed to the MERRA-2 meteorology.

The comparison of the GEOS model simulations for June, July, and August of 2012 shows the baseline case (Fig. 8a) and the case with enhanced dust emission (Fig. 8b). The sensitivity study was performed during this time period because it corresponds

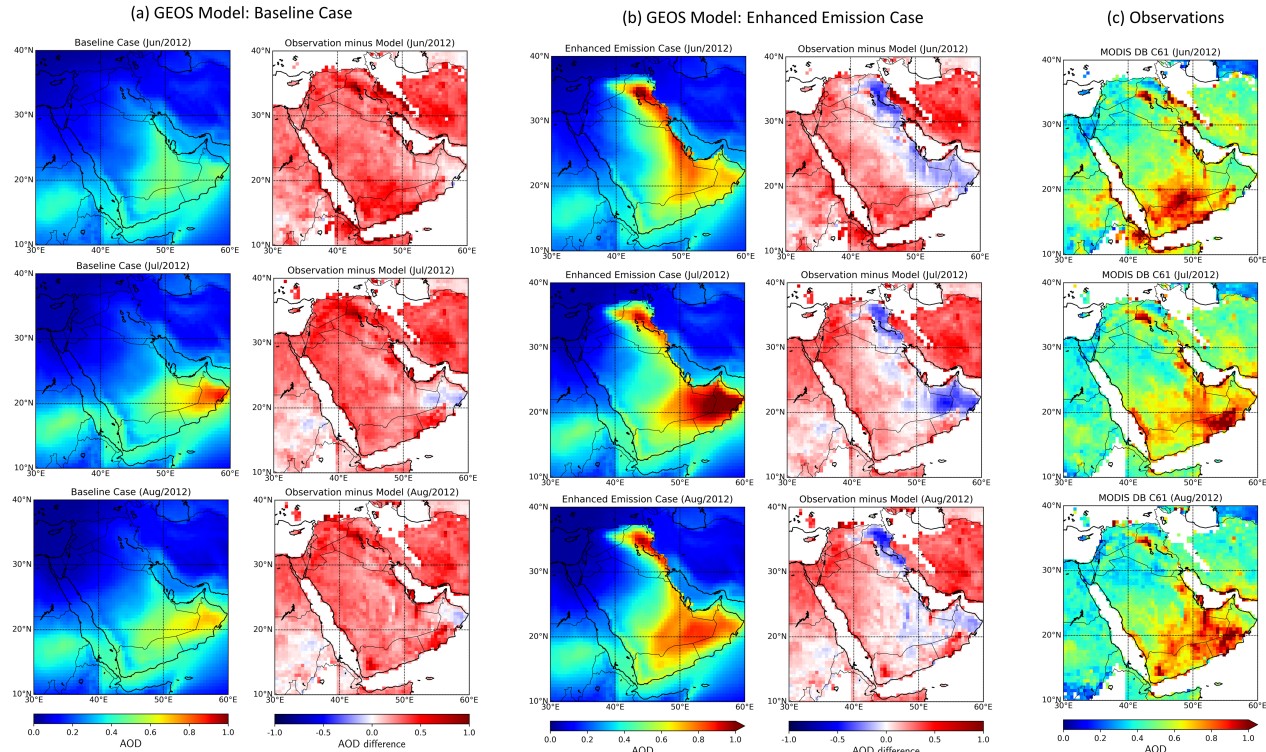

**Figure 8.** Monthly AOD and difference between MODIS DB C6.1 observations and GEOS model simulations for the months of June, July, and August 2012: a) baseline case using the standard topographic source, b) enhanced dust emission sources with optimum matching, and c) MODIS DB C6.1 AOD.

to the highest observed AOD and lowest NDVI values at the Fertile Crescent region. NDVI values over the Fertile Crescent region remained around 0.1 (bare ground) throughout that entire year. However, it is important to note that we did not use the NDVI values in the GEOS simulation. Instead, we allowed more emissions in the grid boxes where the NDVI values changed significantly. The results confirm that the enhancement of dust emissions at the Fertile Crescent region increases dust AOD downwind. The contribution of the dust from the masked region was added to the total AOD. In order to find the optimum enhancement factor, we minimized the difference between MODIS Deep Blue Collection 6.1 monthly mean AODs and the model simulated values for the dust enhanced case in the grid boxes covering the Middle East region. We found that emissions over the Fertile Crescent would have to be enhanced by a factor of 10 for optimum matching with the observations. These enhancement factors were found to be 11.04, 11.26, and 9.27 for June, July, and August of 2012, respectively and are shown in Fig. 8b.

Simulation with enhancement of dust emission in the Fertile Crescent reduced the overall differences between the AOD from model simulation and MODIS observations in downwind regions, specifically in the central and western areas (Fig. 8a-b,

panels to the right). However, they also show AOD can be overestimated regionally near the east coast, which implies that particular attention should be given to regional studies.

## 4   Conclusions

Analysis using observations from the Moderate-resolution Imaging Spectroradiometer (MODIS), Collection 6.1 and the Multi-angle Imaging Spectroradiometer (MISR) shows a positive slope of the deseasonalized AOD (i.e., the AOD with the monthly climatology of the seasonal cycle removed) over an extended region in the Middle East, with a positive significant AOD trend of 0.02-0.04 $\mathrm{yr}^{-1}$ in some areas, during the period between 2003-2012. MISR AOD trends are notably higher than MODIS in the south-east border of Saudi Arabia, Oman, and United Arab Emirates. Ground-based AOD observations from AERONET at Solar Village captured a positive AOD trend of 0.017 $\mathrm{yr}^{-1}$ over the same period.

MERRA-2 Reanalysis (with aerosol data assimilated) captures part of this southern variability, though in smaller magnitude. MERRA-2 GMI Replay (without aerosol data assimilated) is unable to capture most of this variability. Both reanalysis and replay use approximately the same wind fields. The slope of the 10-meter wind speed, which is used to calculate dust emissions, shows statistically significant negative variation over a few small areas in Saudi Arabia, which would have contributed to a regional decrease in dust emissions. Because those locations are not highly efficient emitting areas according to the topographic source function, that effect propagated to dust emission does not impact emission significantly.

Different soil moisture datasets were used in MERRA-2 Reanalysis and MERRA-2 GMI Replay. Most notable positive differences between them are seen over Oman and over a border of Caspian Sea in Iran, where the slope of the surface soil wetness significantly increases in MERRA-2 Reanalysis, but the same is not seen in the slope of the soil wetness used in MERRA-2 GMI Replay. Those positive slopes in soil wetness would have contributed to an overall decrease in dust emission, but again since they are not a major emitting area, the propagated impact in dust emissions is not significant. Finally, both soil moisture datasets indicate slight decrease in soil moisture over the border between Iraq and Iran which, as seen in Fig. 4c, are not translated into significant variation in dust emissions in a manner that would explain the increase in AOD.

It should be noted, however, that even though surface winds and soil moisture in the model do not seem to produce an overall significant effect on dust emissions, both variables from MERRA-2 Reanalysis and MERRA-2 GMI Replay contain errors, and the trends revealed in Fig. 4 have uncertainties. For instance, surface winds in the reanalysis may be underestimated. A comparison of surface winds from reanalysis with high frequency wind observations shows many significant biases, including incorrect annual and seasonal dependences and systematic underestimation of the strongest winds speeds, which can directly impact estimates of dust emission (Largeron et al., 2015; Evan, 2018). Similarly, soil moisture also contains errors since, besides observed precipitation over land, MERRA-2 does not assimilate land surface observations (Reichle et al., 2017a).

The variation of the slope of the AOD analysis increment assimilated in MERRA-2 Reanalysis, similarly to the observations, shows a significant positive AOD increase over the Middle East region, which shows the importance of the aerosol data assimilation and highlights the need to improve the description of the parameters and processes related to the dust properties, emission, and transport in the model. Further investigation with long term and simultaneous observations of meteorological

and surface parameters related to dust emission near dust sources seems to be a key requirement to solve parameterization of dust emission schemes and they are essential to capture dust emission and AOD variability in the models.

The exact causes of the recent AOD increase in the Middle East have not been clearly identified, but within many factors, it has also been linked to droughts and deforestation. Observations show significant change in vegetation coverage in the Fertile Crescent region over the same period (2003-2012). Negative variation in NDVI values of up to 80% in the most critical
areas just to the northwest of the peak in the AOD have supported these findings, as well as the significant increase in the dryness of the soil over the region. The sensitivity study confirmed that enhancing dust emissions in the Fertile Crescent region can reduce the overall differences between the model simulation and MODIS observations comparatively to a baseline case that used the default dust emission scheme based on topographic source. These results support the hypothesis that the increasing AOD observed in the Middle East region could have been associated with land use and reduction of the vegetation
in the region of the Fertile Crescent, which potentially expose more bare soils, increasing dust emissions. However, while the enhancement of the emissions improves AOD in the central and western areas, it does overestimate near the east coast which shows the limitations of this approach to solve regional studies. This work underlines the need for long-term ground-based observations near dust emitting sources to inform models and to track Earth's long-term variability. The correct spatial and temporal description of dust sources are essential to provide the framework needed for accurate determination of local and
global dust loading and transport.

**Appendix A:**

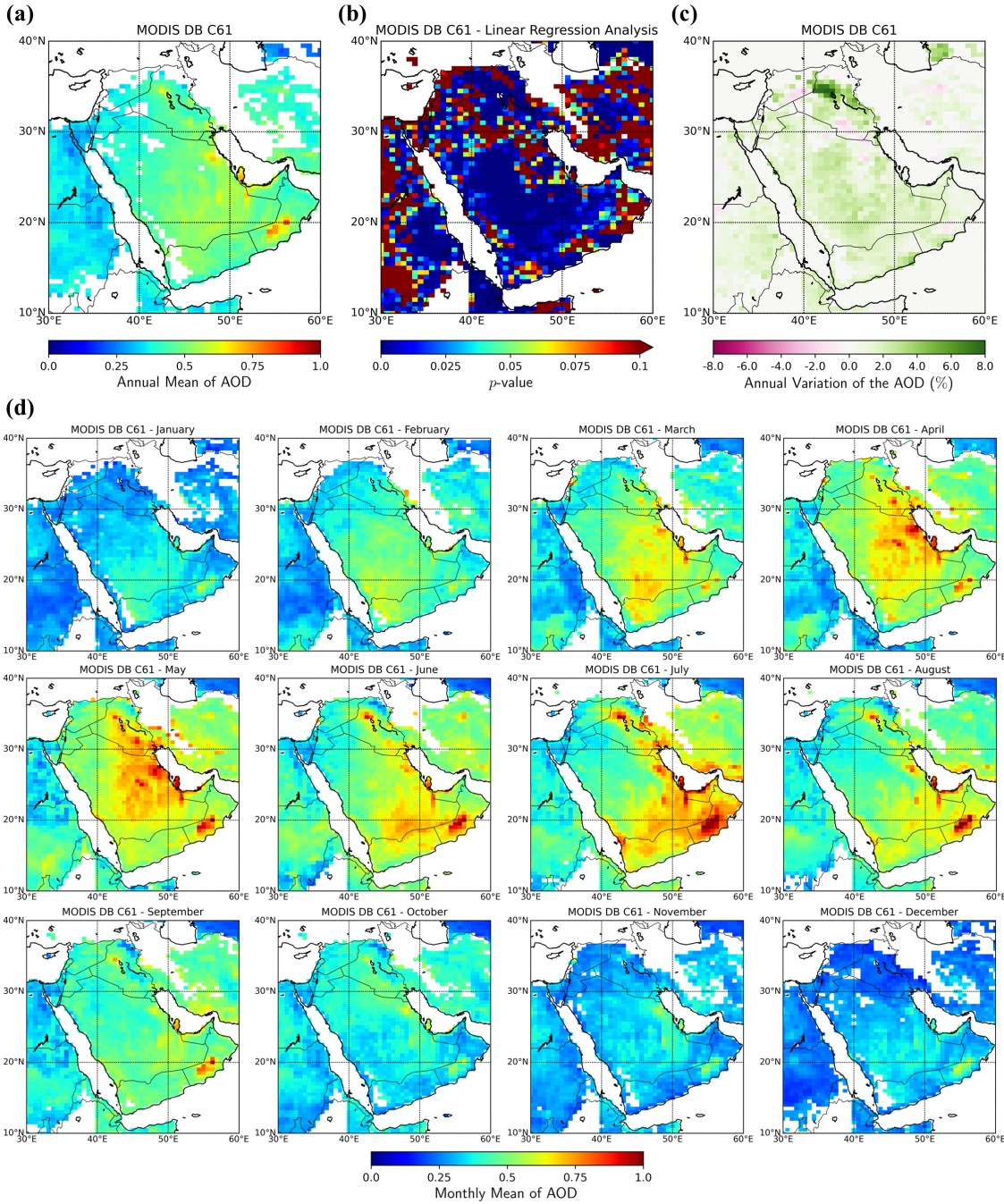

**Figure A1.** AOD from MODIS Deep Blue Collection (6.1) for the period between 2003-2012: a) annual mean, b) *p*-values of the linear regression fitting of the slope, c) annual variation in percentage, and d) monthly means.

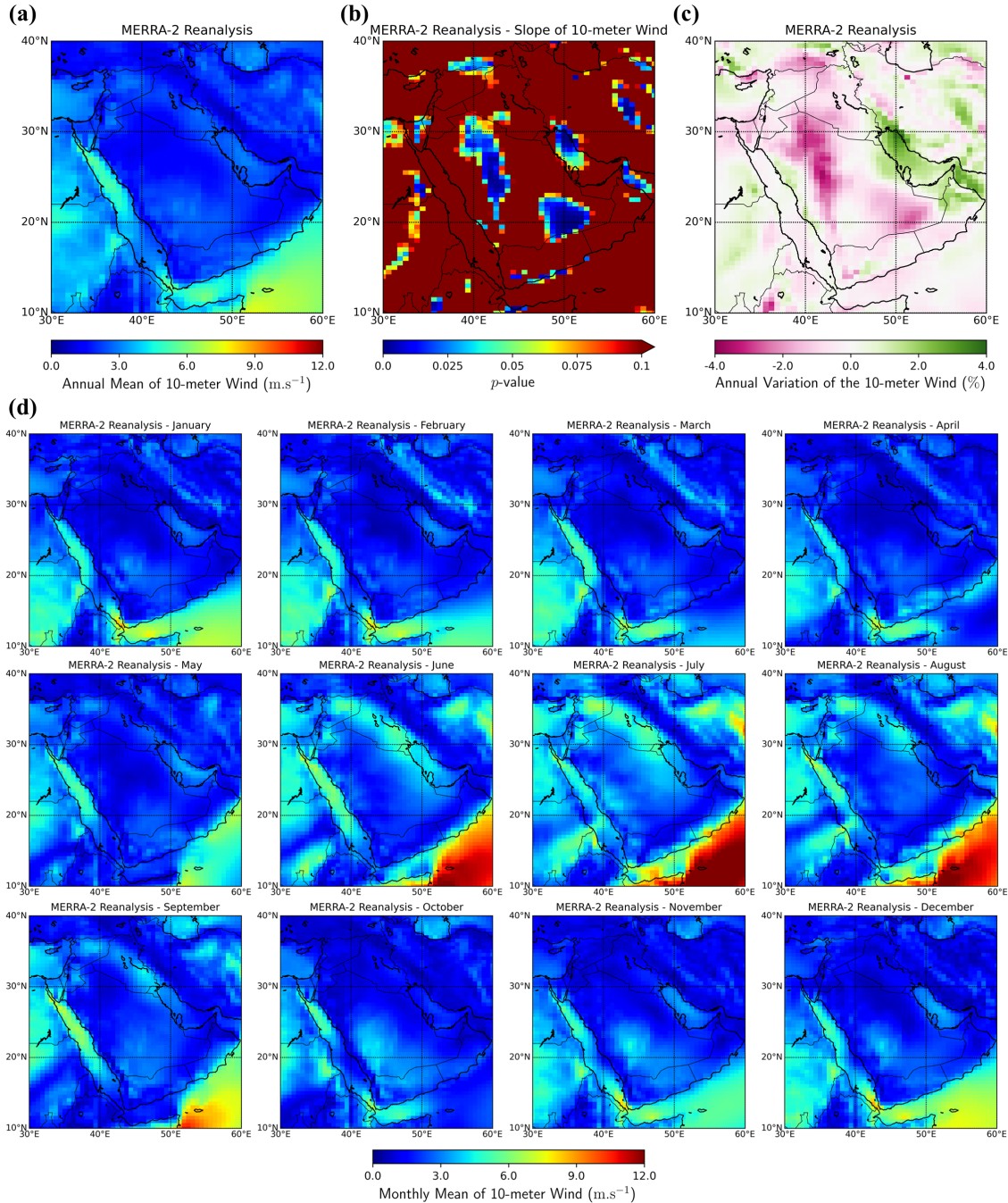

**Figure A2.** MERRA-2 Reanalysis - Wind Speed at 10-meter for the period between 2003-2012: a) annual mean, b) *p*-values of the linear regression fitting of the slope, c) annual variation in percentage, and d) monthly means.

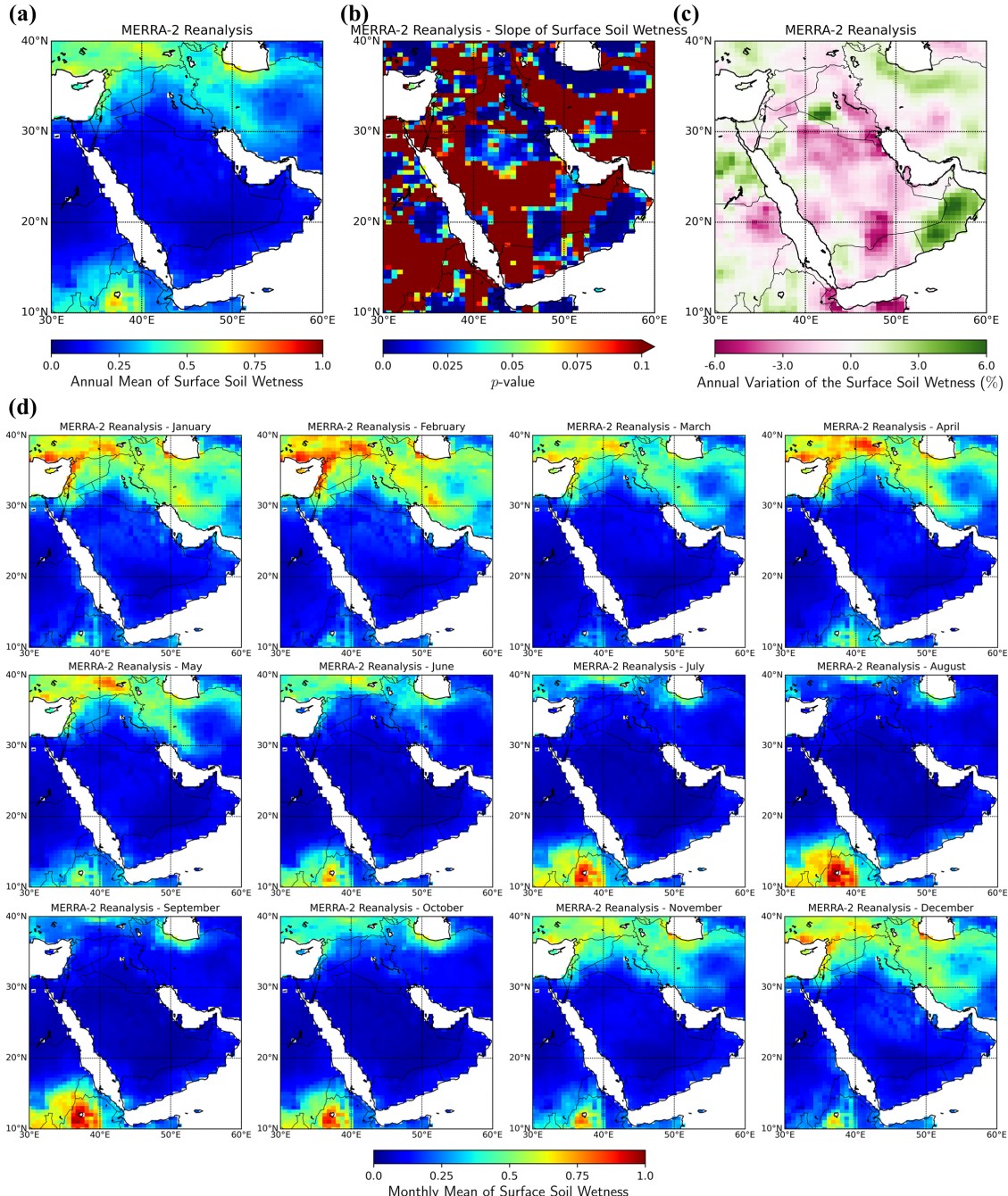

**Figure A3.** MERRA-2 Reanalysis – Surface Soil Wetness for the period between 2003-2012: a) annual mean, b) *p*-values of the linear regression fitting of the slope, c) annual variation in percentage, and d) monthly means.

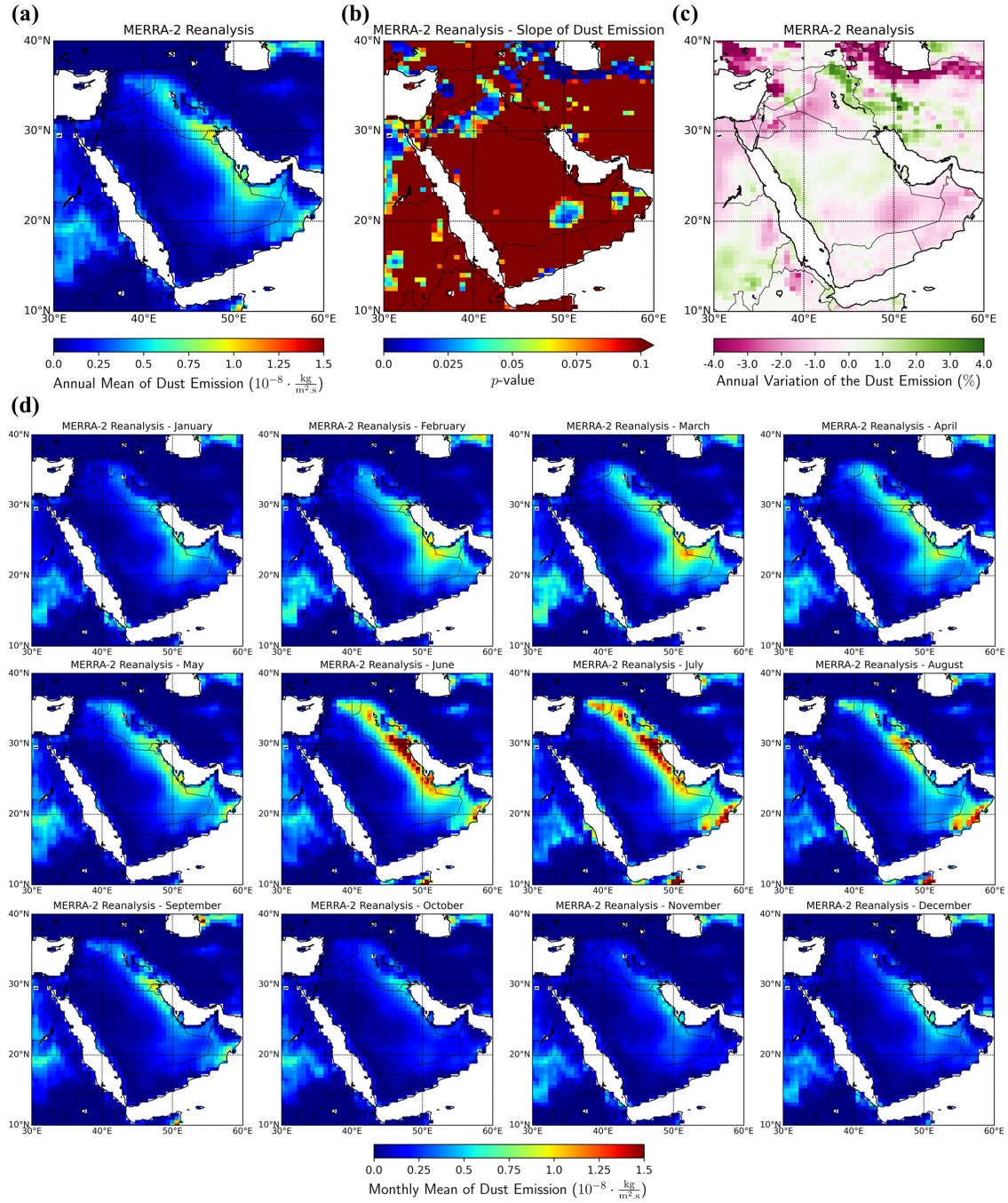

**Figure A4.** MERRA-2 Reanalysis – Dust Emission for the period between 2003-2012: a) annual mean, b) *p*-values of the linear regression fitting of the slope, c) annual variation in percentage, and d) monthly means.

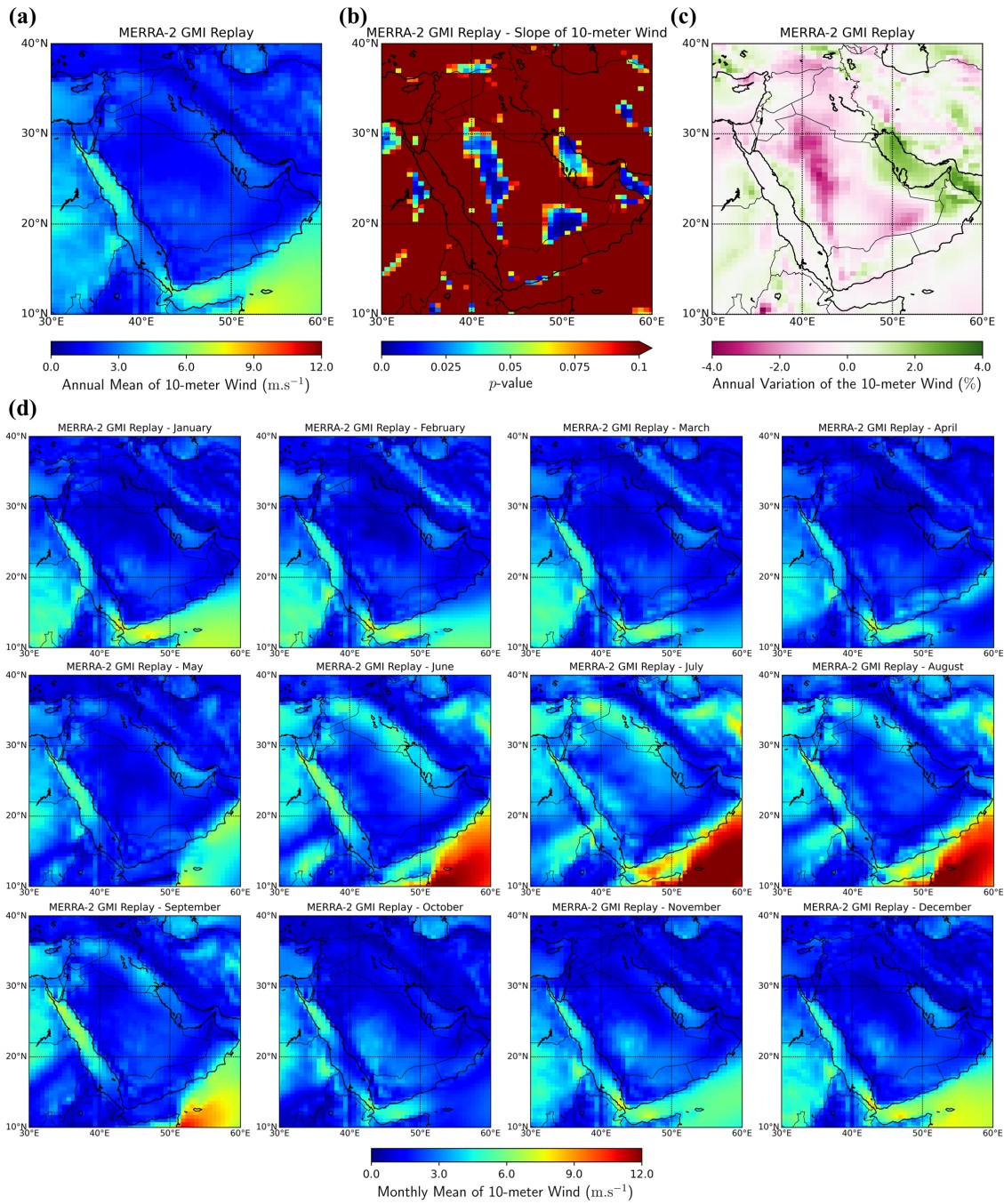

**Figure A5.** MERRA-2 GMI Replay – Wind Speed at 10-meter for the period between 2003-2012: a) annual mean, b) *p*-values of the linear regression fitting of the slope, c) annual variation in percentage, and d) monthly means.

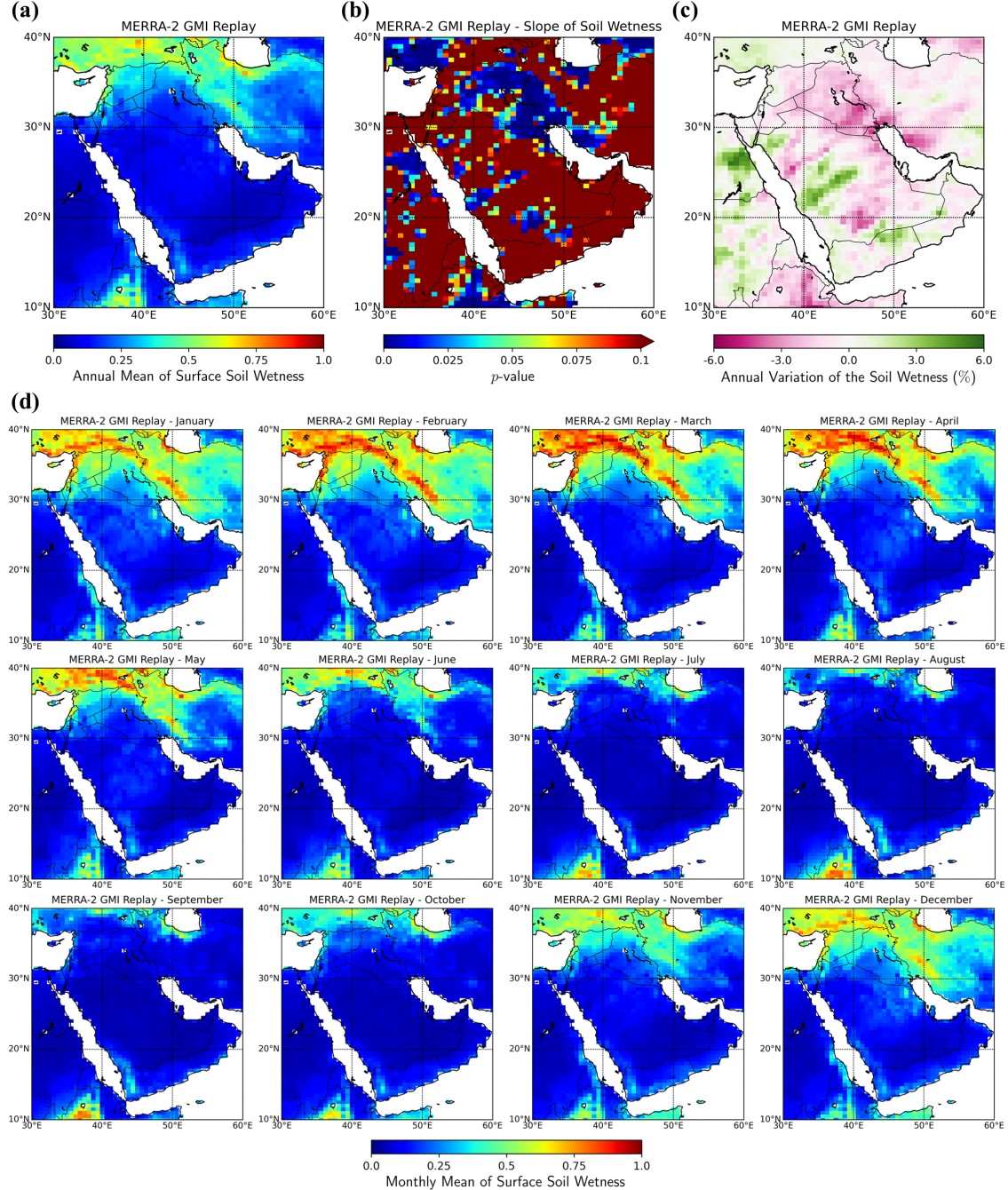

**Figure A6.** MERRA-2 GMI Replay – Surface Soil Wetness for the period between 2003-2012: a) annual mean, b) *p*-values of the linear regression fitting of the slope, c) annual variation in percentage, and d) monthly means.

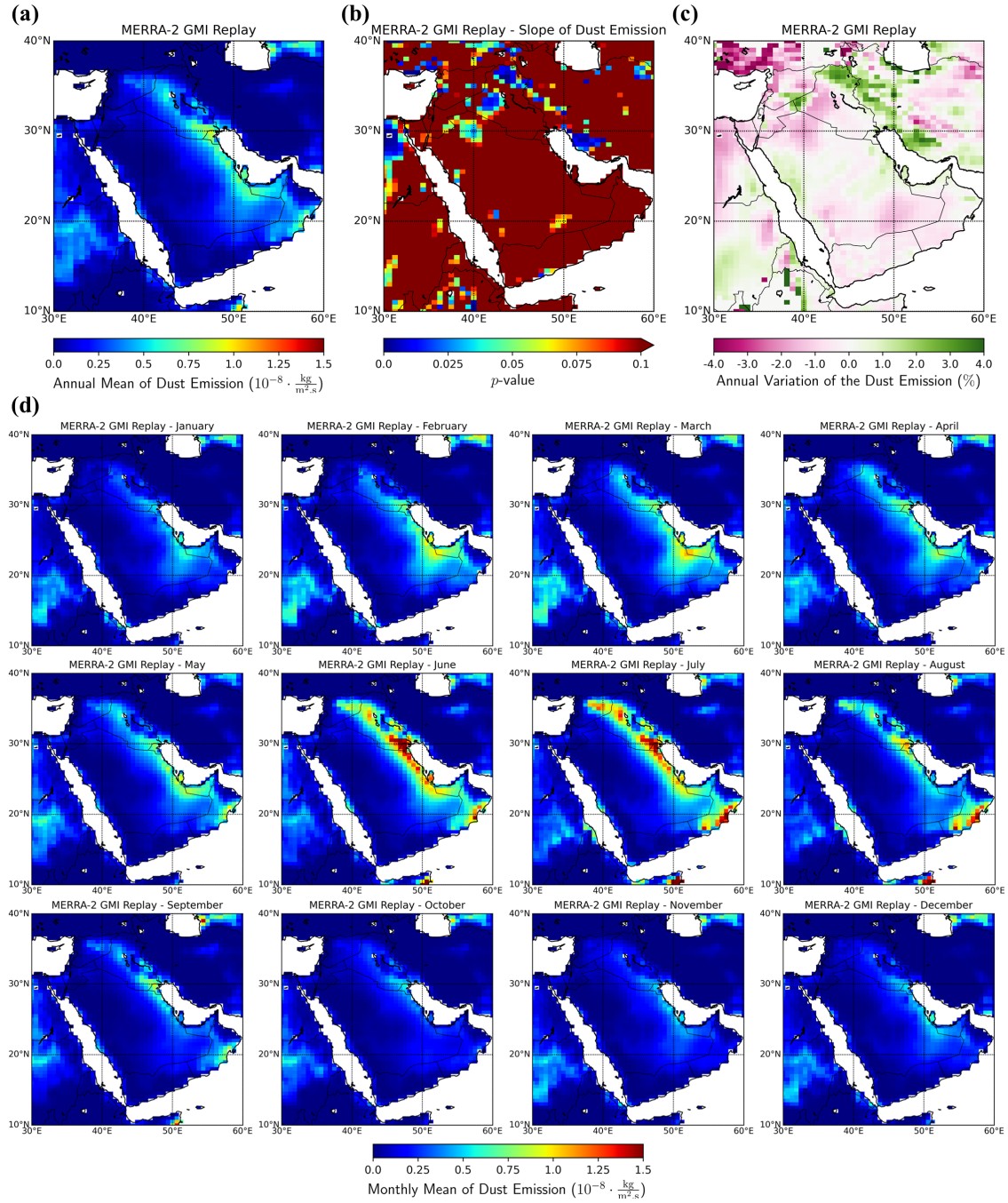

**Figure A7.** MERRA-2 GMI Replay – Dust Emission for the period between 2003-2012: a) annual mean, b) *p*-values of the linear regression fitting of the slope, c) annual variation in percentage, and d) monthly means.

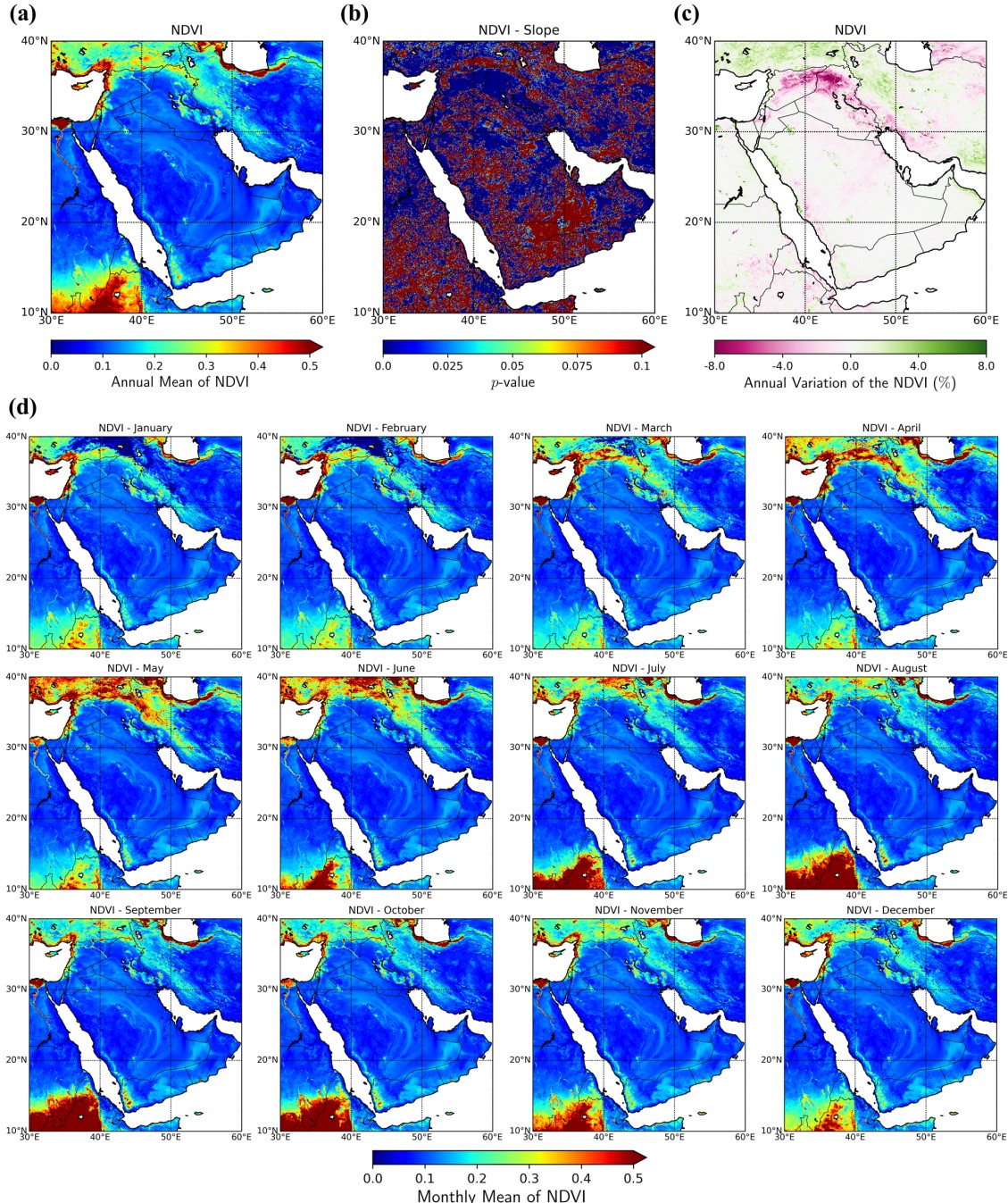

**Figure A8.** Normalized Difference Vegetation Index (NDVI) for the period between 2003-2012: a) annual mean, b) *p*-values of the linear regression fitting of the slope, c) annual variation in percentage, and d) monthly means.

*Data availability.* Aerosol optical depth datasets from MODIS Deep Blue Collection 6.1 and MISR Level 2 product are available at https://www.earthdata.nasa.gov/. AERONET aerosol datasets are available at https://aeronet.gsfc.nasa.gov/. MERRA-2 Reanalysis and MERRA-2 GMI Replay data are available in the following repositories: https://disc.gsfc.nasa.gov/ and http://acd-ext.gsfc.nasa.gov/Projects/-
GEOSCCM/MERRA2GMI/. The Normalized Difference Vegetation Index (NDVI) from MYD13C2 MODIS Version 6 data product is available at: https://lpdaac.usgs.gov/products/myd13c2v006/. Daily FluxSat GPP product can be accessed at: https://avdc.gsfc.nasa.gov/.

*Author contributions.* ARL was responsible for the implementation, scientific analysis, and preparation of this paper. PC made substantial contributions to the concept, design, and interpretation of results presented in this paper. AD provided scientific and technical expertise and assisted with a critical revision of this paper. EN and AS contributed with scientific advice. LO provided expertise on the MERRA-2 GMI
Replay datasets.

*Competing interests.* The authors declare that they have no conflict of interest.

*Acknowledgements.* The authors thank the principal investigators and their teams for making available MODIS, MISR, AERONET, and FluxSat GPP products. This research was funded by the NASA ACMAP project "Assessment of the spatial and temporal variability of mineral dust aerosols in the Middle East and North Africa using observations and modeling"; Project/Grant NNX17AH28G, Richard Eckman and
Kenneth Jucks, program managers. Simulations with the GEOS model were performed at the NASA Center for Climate Simulation (NCCS) computational facility.

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
