# Peer review of "Investigation of observed dust trends over the Middle East region in NASA GEOS Earth system model simulations"

_EGUsphere, 2023_

## Referee Comment (RC1)

**Review of "Investigation of observed dust trends over the Middle East region in NASA GEOS Earth system model simulations" by Adriana Rocha-Lima et al.**

**General comments:**
This paper examines the capability of the NASA GEOS model to reproduce the observed positive AOD trend over the Middle East during 2003–2012. It is found that the model output without aerosol assimilation (MERRA-2 GMI Replay) only shows a weakly positive trend over southern Saudi Arabia and eastern Oman, while the simulation with aerosol assimilation (MERRA-2) largely captures the spatial pattern of the AOD trend shown in satellite retrievals, although in a weaker magnitude and the hot spots over the central to eastern Fertile Crescent region are missing. A sensitivity test that allows dust emissions over the regions with a strong decreasing trend in NDVI increases AOD in western Syria and Iraq and downwind regions over southeastern Saudi Arabia and Oman, largely reducing the discrepancies between model output and MODIS AOD. This suggested that vegetation reduction in the Fertile Crescent region contributes to the observed positive trend of AOD. While the findings advance the understanding of the model's capability to capture long-term dust trends in the Middle East and the role of vegetation in dust trends in the region, some details about simulation settings and the selection of analyzed region and months need further clarification.

**Specific comments/suggestions:**
1. Section 1.1 reviews previous observational and modeling studies of AOD over the Middle East. However, it is not very clear why the study focuses on 2003–2012 instead of a longer period, i.e., 2003 to the present. Are there any trends in AOD from 2013 to 2022 (or 2023)? I think it's informative to place the current study of the positive trend in AOD in the context of long-term variations in AOD in the region. For instance, previous studies (e.g., Notaro et al. 2015) related AOD variability in the Fertile Crescent to low-frequency variations of the Pacific Decadal Oscillation. Is this positive trend of AOD during 2003–2012 part of long-term decadal variations in AOD in the Middle East?

2. Section 2, it would be more informative to add some comparisons with previous studies when discussing findings in Sections 2-3.

3. It's helpful to add a data section to briefly introduce the satellite products (e.g., MODIS and MISR AOD, MODIS NDVI) used in the study.

4. The dust source function in the GOCART module is determined by both topographical depression and surface bareness so dust aerosols are emitted from bare ground or sparsely vegetated regions (Kim et al. 2013). I wonder if an NDVI threshold/mask is applied to the source map shown in Fig. 7a in the baseline simulation as well. And if so, is it a climatological mean or time-varying NDVI?

5. The setting of the sensitivity test (lines 201-206) needs a bit more clarification. For instance, a region with a strong decreasing trend in NDVI is selected (i.e., $<-0.0025 \text{ yr}^{-1}$) to allow dust emissions. However, it is not clear what the absolute value of NDVI is in the selected region and whether NDVI in the region is low (i.e., sparsely vegetated region or bare ground) all the time during the simulation. It is also not clear how long the simulation is conducted.

As shown in Fig. 8, AOD in the masked area is higher in the dust-enhanced case than in the baseline case, which indicates that dust emissions in the same area are in fact suppressed in the baseline simulation. Is this correct?

6. It is not fully clear why the comparisons between model results and MODIS focus on JJA. Is it the time when NDVI shows the strongest decreasing trend or when MODIS AOD shows the greatest trend? Since the earlier discussion uses monthly data or annual values, it is better to add some discussion to justify the selection.

7. Fig. 4 suggested that both soil moisture and surface wind speeds play little role in the increasing trend of AOD in the model. It should be noted both variables from MERRA-2 or MERRA-2 GMI Replay may contain errors from the model thus the trends revealed in Fig. 4 have uncertainties. For instance, surface winds in the reanalysis may be underestimated (e.g., Largeron et al. 2015; Evan et al. 2018). It would be interesting to compare the trends found there with studies using ground observations.

8. While many previous studies of AOD or dust trends are discussed in section 1.1, a few recent papers also examined aerosol trends in the Middle East, e.g., Chen et al. 2019; Song et al. 2021; Xi 2021; Sabetghadam et al. 2021; Liu et al. 2023.

9. Line 93, "constrained by the MERRA-2 Reanalysis", can you please provide more details about the constraints? Are MERRA-2 meteorological fields prescribed?

10. Line 97, "prescribed soil", from what dataset?

11. Lines 100-103, please provide temporal resolution of the datasets.

12. Line 189, please add a couple of lines to introduce the FluxSat GPP data.

13. line 213, are the results from the optimum-matching run shown in Fig. 8?

14. Fig. 8, in addition to comparing AOD patterns, have you examined the trend of AOD from the dust-enhanced case? Are the simulated magnitude and pattern more consistent with MODIS than the baseline simulation?

15. Line 227, "use the same wind fields", Fig. 4 shows wind fields in the two datasets are slightly different from each other.

16. Line 245, "it has also been linked to deforestation", the reduction in NDVI could be due to both droughts and deforestation.

**Technical corrections**
Fig. 1 caption, please explain the red shading in 1(b)
Fig. 4, do white contours denote areas with a p-value less than 0.05? If so, please add the info to the figure caption. It's somewhat redundant to show maps of p-values if the contours of significant areas are overlayed on the regression slopes.

*References:*

Che, H., Gui, K., Xia, X., Wang, Y., Holben, B. N., Goloub, P., Cuevas-Agulló, E., Wang, H., Zheng, Y., Zhao, H., and Zhang, X.: Large contribution of meteorological factors to inter-decadal changes in regional aerosol optical depth, Atmos. Chem. Phys., 19, 10497–10523, https://doi.org/10.5194/acp-19-10497-2019, 2019.

Evan, A. T.: Surface winds and dust biases in climate models. Geophysical Research Letters, 45, 1079–1085. https://doi.org/10.1002/2017GL076353, 2018.

Kim, D., M. Chin, H. Bian, Q. Tan, M. E. Brown, T. Zheng, R. You, T. Diehl, P. Ginoux, and T. Kucsera: The effect of the dynamic surface bareness on dust source function, emission, and distribution, J. Geophys. Res. Atmos., 118, 871–886, doi: 10.1029/2012JD017907, 2013.

Largeron, Y., Guichard, F., Bouniol, D., Couvreux, F., Kergoat, L., and Marticorena, B. : Can we use surface wind fields from meteorological reanalyses for Sahelian dust emission simulations?, Geophys. Res. Lett., 42, 2490–2499, https://doi.org/10.1002/2014gl062938, 2015.

Liu, G. Li, J. and Ying, T.: The shift of decadal trend in Middle East dust activities attributed to North Tropical Atlantic variability, Science Bulletin, 68 (2023), 1439–1446, https://doi.org/10.1016/j.scib.2023.05.031, 2023.

Sabetghadam, S., Alzadeh, O., Khoshsima, M., and Pierleoni, A.: Aerosol properties, trends and classification of key types over the Middle East from satellite-derived atmospheric optical data, Atmospheric Environment, 246 (2021) 118100, https://doi.org/10.1016/j.atmosenv.2020.118100, 2021.

Xi, X. : Revisiting the recent dust trends and climate drivers using horizontal visibility and present weather observations. Journal of Geophysical Research: Atmospheres, 126, e2021JD034687, 2021.

---

## Author Comment (AC1)

**Review of "Investigation of observed dust trends over the Middle East region in NASA GEOS Earth system model simulations" by Adriana Rocha-Lima et al.**

**General comments:**
This paper examines the capability of the NASA GEOS model to reproduce the observed positive AOD trend over the Middle East during 2003–2012. It is found that the model output without aerosol assimilation (MERRA-2 GMI Replay) only shows a weakly positive trend over southern Saudi Arabia and eastern Oman, while the simulation with aerosol assimilation (MERRA-2) largely captures the spatial pattern of the AOD trend shown in satellite retrievals, although in a weaker magnitude and the hot spots over the central to eastern Fertile Crescent region are missing. A sensitivity test that allows dust emissions over the regions with a strong decreasing trend in NDVI increases AOD in western Syria and Iraq and downwind regions over southeastern Saudi Arabia and Oman, largely reducing the discrepancies between model output and MODIS AOD. This suggested that vegetation reduction in the Fertile Crescent region contributes to the observed positive trend of AOD. While the findings advance the understanding of the model's capability to capture long-term dust trends in the Middle East and the role of vegetation in dust trends in the region, some details about simulation settings and the selection of analyzed region and months need further clarification.

The authors would like to thank the anonymous Referee #2 for the detailed and insightful review of this manuscript.

**Specific comments/suggestions:**
1. Section 1.1 reviews previous observational and modeling studies of AOD over the Middle East. However, it is not very clear why the study focuses on 2003–2012 instead of a longer period, i.e., 2003 to the present. Are there any trends in AOD from 2013 to 2022 (or 2023)? I think it's informative to place the current study of the positive trend in AOD in the context of long-term variations in AOD in the region. For instance, previous studies (e.g., Notaro et al. 2015) related AOD variability in the Fertile Crescent to low-frequency variations of the Pacific Decadal Oscillation. Is this positive trend of AOD during 2003–2012 part of long-term decadal variations in AOD in the Middle East?

Response: The main reason we focused this study on the period between 2003-2012 is that ground-based and satellite observations indicate significant positive AOD trends over this period. After that, different studies show that the trend was reversed (Klingmuller et al. (2016), Notaro et al. (2015)). We have not evaluated the AOD trends in the model beyond 2013, but we agree that this in an interesting question and a natural continuation of this study.
In relation to the second question, we have now included in the introduction more recent studies that investigate the link between dust activity in the Middle East and climate decadal oscillations. In Liu et al. 2023, the authors related the shift of the AOD trend in the Middle East from positive to negative to the variability in the North Tropical Atlantic (NTA) sea surface temperature (SST)

around 2010. In Xi (2021), the authors associated the reduction in dust activity after 2015 with a shift in climate towards more El Niño-like conditions and a combination of positive and weak negative Pacific Decadal Oscillations (PDO). In principle, the model meteorology would respond to those forcings to the extent they are captured in the model, however, as shown in Figure 4, trends in 10-meter winds and surface soil wetness in the same period do not produce a significant increase in dust emissions. This also means that the impact of long-term decadal variations in the model's dust emission scheme is not totally comprehended.

We have added the following paragraph in the manuscript to include these points:

*"Conversely, Che et al. (2019) found that sea level pressure and wind speed were the primary meteorological factors driving AOD variations over the Middle East. More recent studies have examined the link between dust activity in the Middle East and climate decadal oscillations. Xi (2021) associated the AOD trends over the Middle East with the combined effects from El Niño-Southern Oscillation (ENSO) and the Pacific Decadal Oscillations (PDO). Specifically, when both ENSO and PDO are in phase, influences in the sea surface temperature and winds are amplified, creating high surface pressure around the Middle East with hotter and drier conditions. The drought in the Tigris-Euphrates basin is believed to be associated with effects of La Niña and negative PDO phases, which can have resulting effects for agriculture and vegetation loss in the Fertile Crescent. At the same time, Liu et al. (2023) related the shift of the AOD trend in the Middle East around 2010 from positive to negative to the shift in the North Tropical Atlantic (NTA) Sea Surface Temperature (SST)."*

2. Section 2, it would be more informative to add some comparisons with previous studies when discussing findings in Sections 2-3.

Response: For clarity, we would like to have these sections focused on the comparison between the model simulations MERRA-2 Reanalysis and MERRA-2 GMI Replay, and satellite observations.

3. It's helpful to add a data section to briefly introduce the satellite products (e.g., MODIS and MISR AOD, MODIS NDVI) used in the study.

Response: We added a table in section 2 summarizing the satellite products used in this study The table added in section 2 is showed below.

*"Table 1 summarizes ground-based and satellite products used in this study. All datasets were selected to cover the period between 2003 and 2012."*

**Table 1.** List of sensors and dataset products used in this study in the period from 2003 to 2012.

| Sensor | Dataset Description |
|---|---|
| AERONET Sun-photometer | AOD Level 2.0, Solar Village (24.9N, 46.4E) (Holben et al., 1998) |
| MISR Terra Satellite | AOD Level 2 Aerosol (NASA/LARC/SD/ASDC, 1999) |
| MODIS Terra Satellite | AOD Deep Blue Collection 6.1 (Hsu et al., 2019; Sayer et al., 2019) |
| MODIS Aqua Satellite | Normalized Difference Vegetation Index (NDVI) MYD13C2 Version 6, Level 3 product, 0.05 degree (Didan, 2015) |
| MODIS Terra/Aqua Satellite | Global Daily Terrestrial Gross Primary Production (FluxSat GPP) Version 2.0 (Joiner and Yoshida, 2020) |

4. The dust source function in the GOCART module is determined by both topographical depression and surface bareness so dust aerosols are emitted from bare ground or sparsely vegetated regions (Kim et al. 2013). I wonder if an NDVI threshold/mask is applied to the source map shown in Fig. 7a in the baseline simulation as well. And if so, is it a climatological mean or time-varying NDVI?

Response:  The default configuration of the dust emission scheme in GOCART used in MERRA-2, MERRA-2 GMI, and in the sensitivity study simulations included on the manuscript is the static topographic depression source map (Ginoux et al, 2001) and does not incorporate a time-varying NDVI. This information is mentioned in (Line: 214, Section 3.2). We have now included the following paragraph to add Kim's study and to clarify this information.

*"A study by Kim et al. (2013) showed that the time dependence of global dust sources can have significant impacts on dust simulations near source regions. However, the static topographic source function used in MERRA-2 Reanalysis and MERRA-2 GMI Replay does not incorporate time-varying NDVI. To assess the impact of the variation of vegetation on dust emissions, we performed a GEOS simulation (baseline case) using the default static topographic depression source map (Fig. 7a) and a second simulation with a modified source (dust enhanced case)."*

5. The setting of the sensitivity test (lines 201-206) needs a bit more clarification. For instance, a region with a strong decreasing trend in NDVI is selected (i.e., $<-0.0025$ yr$^{-1}$) to allow dust emissions. However, it is not clear what the absolute value of NDVI is in the selected region and whether NDVI in the region is low (i.e., sparsely vegetated region or bare ground) all the time during the simulation.  It is also not clear how long the simulation is conducted.

Response: To illustrate the magnitude of the NDVI, we show the absolute monthly values of NDVI for a grid box in the region of the Fertile Crescent in Fig. 6a. We found that NDVI for that grid box ranged from around 0.1 (bare ground) to 0.4 (sparsely vegetated) at different times of the year. To provide a better understanding of the NDVI values for the entire region, we have now added maps of the NDVI values in Fig. A8. The sensitivity study simulations were performed only for JJA in 2012, and NDVI values over the Fertile Crescent region remained around 0.1(bare ground) throughout that entire year. However, it is important to note that we did not use the NDVI values in the GEOS simulation. Instead, we allowed for more emissions in the

grid boxes where the NDVI values significantly changed.  The following sentence has been added to clarify these points:

*"The greenest peak in March-April is suppressed over the years, with NDVI values ranging from 0.4 (sparsely vegetated) in the year 2003 to 0.1 (bare ground) in 2012. NDVI maps for the entire region are shown in Fig. A8."*

As shown in Fig. 8, AOD in the masked area is higher in the dust-enhanced case than in the baseline case, which indicates that dust emissions in the same area are in fact suppressed in the baseline simulation. Is this correct?

Response. There is no suppression in the dust emissions in the baseline case. In other words, the "baseline case" is a standard model run with the default static topographic source values showed in Fig. 7a.

6. It is not fully clear why the comparisons between model results and MODIS focus on JJA.  Is it the time when NDVI shows the strongest decreasing trend or when MODIS AOD shows the greatest trend? Since the earlier discussion uses monthly data or annual values, it is better to add some discussion to justify the selection.

Response: Yes, we have included the following paragraph to provide justification for our choice:

*"The sensitivity study was performed during this time period because it corresponds to the highest observed AOD and lowest NDVI values at the Fertile Crescent region. NDVI values over the Fertile Crescent region remained around 0.1(bare ground) throughout that entire year. However, it is important to note that we did not use the NDVI values in the GEOS simulation. Instead, we allowed more emissions in the grid boxes where the NDVI values changed significantly."*

7. Fig. 4 suggested that both soil moisture and surface wind speeds play little role in the increasing trend of AOD in the model. It should be noted both variables from MERRA-2 or MERRA-2 GMI Replay may contain errors from the model thus the trends revealed in Fig. 4 have uncertainties. For instance, surface winds in the reanalysis may be underestimated (e.g., Largeron et al. 2015; Evan et al. 2018). It would be interesting to compare the trends found there with studies using ground observations.
Response: That is great point. We have added the following paragraph to include this information. Additionally, a manuscript comparing the 10-meter wind speed with ground-based observation for the Middle East region is currently being prepared by E. Faber et al..

*"It should be noted, however, that even though surface winds and soil moisture in the model do not seem to produce an overall significant effect on dust emissions, both variables from MERRA-2 Reanalysis and MERRA-2 GMI Replay contain errors, and the trends revealed in Fig. 4 have uncertainties. For instance, surface winds in the reanalysis may be underestimated. A*

*comparison of surface winds from reanalysis with high frequency wind observations shows many significant biases, including incorrect annual and seasonal dependences and systematic underestimation of the strongest winds speeds, which can directly impact estimates of dust emission (Largeron et al., 2015; Evan et al., 2018). Similarly, soil moisture also contains errors since, besides observed precipitation over land, MERRA-2 does not assimilate land surface observations (Reichle et al., 2017a)."*

8. While many previous studies of AOD or dust trends are discussed in section 1.1, a few recent papers also examined aerosol trends in the Middle East, e.g., Chen et al. 2019; Song et al. 2021; Xi 2021; Sabetghadam et al. 2021; Liu et al. 2023.

Response: We have incorporated the suggested references in the introduction (see the response to question 1). Additionally, we have added the following paragraphs to include these studies.

*"According to Hamidi et al. (2013), dust activities in the years preceding 2013 were intensified due to several reasons. These include the development of dam construction projects on the Tigris and Euphrates rivers, which decreased the water content of soil in the downstream areas, urbanization in regions previously used for agriculture, and a shortage of power that hindered the adequate irrigation of farmlands."*

*"Similarly, Sabetghadam et al. (2021) noticed a higher presence of aerosol dust type in March 2012 compared to the long-term average for the period of 2001-2019 in the Fertile Crescent region."*

9. Line 93, "constrained by the MERRA-2 Reanalysis", can you please provide more details about the constraints? Are MERRA-2 meteorological fields prescribed?

Response: MERRA-2 Reanalysis meteorological fields (atmospheric temperature, humidity, and winds, among others) are assimilated. Additionally, observations of precipitation are used to correct model-generated precipitation (Reichle et al., 2017). We have modified this paragraph to provide corrections and clarify this information. Here is the revised paragraph:

*"Two model GEOS simulations were used in this study: MERRA-2 Reanalysis and MERRA-2 GMI Replay. MERRA-2 Reanalysis is a long-term (1980-present) global reanalysis that assimilates satellite meteorological and aerosol data (Gelaro et al., 2017; Randles et al., 2017; Buchard et al., 2017). It assimilates several wind observations, including ground-based datasets, remotely sensed profilers, and satellite derived and retrieved winds (McCarty et al., 2016). MERRA-2 Reanalysis also uses precipitation observations to correct model-generated precipitation, which is needed for estimating soil moisture in the catchment land surface model (De Lannoy et al., 2014, Gelaro et al., 2017, Reichle et al., 2017a, b). Furthermore, MERRA-2 Reanalysis assimilates total AOD from multiple systems, such as the Advanced Very-High-Resolution Radiometer (AVHRR), MODIS, Multi-angle Imaging Spectroradiometer (MISR) over bright surfaces, and from selected AERONET stations prior to 2015. The aerosol assimilation is*

*performed eight times a day at three-hour intervals. MERRA-2 GMI Replay (Strode et al., 2019) uses a replay mechanism to produce a simulation with meteorology similar to MERRA-2 Reanalysis. There are several differences between MERRA-2 GMI Replay and MERRA-2 Reanalysis to note. First, unlike MERRA-2 Reanalysis, MERRA-2 GMI Replay was performed with a full chemistry simulation using the Global Modeling Initiative's (GMI) chemical mechanism (Duncan et al., 2007; Strahan et al., 2007). This has no practical impact on the simulations of dust emissions and loss processes. Second, MERRA-2 GMI Replay, uses the same catchment land surface model as MERRA-2 Reanalysis, however, there are differences in soil moisture due to differences in the treatment of water vapor and precipitation. Finally, and most significantly, MERRA-2 GMI Replay does not constrain the total aerosol optical depth to observations like in MERRA-2 Reanalysis".*

10. Line 97, "prescribed soil", from what dataset?

Response: The information has been corrected in the response above. Soil moisture in MERRA-2 GMI Replay (as well as surface soil wetness) is diagnosed by the catchment land surface model. A detailed description of this model, including the soil properties used, is provided in De Lannoy et al. (2014).

11. Lines 100-103, please provide temporal resolution of the datasets.
Response: We added temporal resolution in the text. The sentence now read as:

"Output is saved **hourly** on a regular grid with resolution of 0.625º longitude and 0.5º latitude."

12. Line 189, please add a couple of lines to introduce the FluxSat GPP data.
Response: We added the following paragraph with more details about the FluxSat GPP data.

*"The Gross Primary Production (GPP), which represents the amount of carbon dioxide (CO2) assimilated by plants through photosynthesis, is an important indicator of vegetation. The GPP product archived at the NASA Aura Validation Data Center (AVDC) was derived using neural networks combined with Bidirectional Reflectance Distribution Function (BRDF) and Nadir Adjusted Reflectance (NBAR) products from the MODIS instrument. It also incorporates global GPP estimates from selected FLUXNET2015 eddy covariance tower sites (Joiner and Yoshida, 2020)".*

13. line 213, are the results from the optimum-matching run shown in Fig. 8?
Response: Yes, the results from the optimum matching are shown in Fig 8b. We have added the wording "with optimum matching" in the caption of Fig. 8 to make this information clear.

14. Fig. 8, in addition to comparing AOD patterns, have you examined the trend of AOD from the dust-enhanced case? Are the simulated magnitude and pattern more consistent with MODIS than the baseline simulation?

Response: We did not compare the AOD trend because the sensitivity study was carried out for the summer (JJA) of 2012. For an adequate comparison of the AOD trend, a much longer simulation would be necessary covering the entire period between 2003-2012.

15. Line 227, "use the same wind fields", Fig. 4 shows wind fields in the two datasets are slightly different from each other.
Response: That is a good point. We have updated the text to be more precise:

"*Both reanalysis and replay use **approximately** the same wind fields.*"

16. Line 245, "it has also been linked to deforestation", the reduction in NDVI could be due to both droughts and deforestation.

Response: We have included that information in the text. The sentence was rewritten as:

"*The exact causes of the recent AOD increase in the Middle East have not been clearly identified, but within many factors, it has also been linked to **droughts and** deforestation.*"

**Technical corrections**
Fig. 1 caption, please explain the red shading in 1(b).
Response: Completed. We included this information in the caption.

"*Red shading corresponds to the Fertile Crescent region.*".

Fig. 4, do white contours denote areas with a p-value less than 0.05? If so, please add the info to the figure caption. It's somewhat redundant to show maps of p-values if the contours of significant areas are overlayed on the regression slopes.
Response: Completed. Yes, we included this information in the caption of the Figure 4. We agree that there is a little bit of redundance in keeping the p-value maps, but we decided to keep them as we think it facilitates the visualization and comparison between plots.

Figure 4: "***White contours** regions with p-values lower than 0.05 are associated with statistically significant slopes*".

Finally, Figure 1a) and Figure 6a) were modified to make them colorblind friendly.

*References:*
Che, H., Gui, K., Xia, X., Wang, Y., Holben, B. N., Goloub, P., Cuevas-Agulló, E., Wang, H., Zheng, Y., Zhao, H., and Zhang, X.: Large contribution of meteorological factors to inter-decadal changes in regional aerosol optical depth, Atmos. Chem. Phys., 19, 10497–10523, https://doi.org/10.5194/acp-19-10497-2019, 2019.

Evan, A. T.: Surface winds and dust biases in climate models. Geophysical Research Letters, 45, 1079–1085. https://doi.org/10.1002/2017GL076353, 2018.

Kim, D., M. Chin, H. Bian, Q. Tan, M. E. Brown, T. Zheng, R. You, T. Diehl, P. Ginoux, and T. Kucsera: The effect of the dynamic surface bareness on dust source function, emission, and distribution, J. Geophys. Res. Atmos., 118, 871–886, doi: 10.1029/2012JD017907, 2013.

Largeron, Y., Guichard, F., Bouniol, D., Couvreux, F., Kergoat, L., and Marticorena, B. : Can we use surface wind fields from meteorological reanalyses for Sahelian dust emission simulations?, Geophys. Res. Lett., 42, 2490–2499, https://doi.org/10.1002/2014gl062938, 2015.

Liu, G. Li, J. and Ying, T.: The shift of decadal trend in Middle East dust activities attributed to North Tropical Atlantic variability, Science Bulletin, 68 (2023), 1439–1446, https://doi.org/10.1016/j.scib.2023.05.031, 2023.

Sabetghadam, S., Alzadeh, O., Khoshsima, M., and Pierleoni, A.: Aerosol properties, trends and classification of key types over the Middle East from satellite-derived atmospheric optical data, Atmospheric Environment, 246 (2021) 118100, https://doi.org/10.1016/j.atmosenv.2020.118100, 2021.

Xi, X. : Revisiting the recent dust trends and climate drivers using horizontal visibility and present weather observations. Journal of Geophysical Research: Atmospheres, 126, e2021JD034687, 2021.